# Intensification characteristics of hydroclimatic extremes in the Asian monsoon region under 1.5 and 2.0 °C of global warming

Jeong-Bae Kim[1], Deg-Hyo Bae[1]

[1] Department of Civil and Environmental Engineering, Sejong University, Seoul, 05006, Korea

*Correspondence to*: Deg-Hyo Bae (dhbae@sejong.ac.kr)

**Abstract.** Understanding the influence of global warming on regional hydroclimatic extremes is challenging. To reduce the potential risk of extremes under future climate states, assessing the change in extreme climate events is important, especially in Asia, due to spatial variability of climate and its seasonal variability. Here, the changes in hydroclimatic extremes are assessed over the Asian monsoon region under global mean temperature warming targets of 1.5 and 2.0 °C above preindustrial

levels based on representative concentration pathways (RCPs) 4.5 and 8.5. Analyses of the subregions classified using regional climate characteristics are performed based on the multimodel ensemble mean (MME) of five bias-corrected global climate models (GCMs). For runoff extremes, the hydrologic responses to 1.5 and 2.0 °C global warming targets are simulated based on the variable infiltration capacity (VIC) model. Changes in temperature extremes show increasing warm extremes and decreasing cold extremes in all climate zones with strong robustness under global warming conditions. However, the hottest

extreme temperatures occur more frequently in low-latitude regions with tropical climates. Changes in mean annual precipitation and mean annual runoff and low runoff extremes represent the large spatial variations with weak robustness based on intermodel agreements. Global warming is expected to consistently intensify maximum extreme precipitation events (usually exceeding a 10 % increase in intensity under 2.0 °C of warming) in all climate zones. The precipitation change patterns directly contribute to the spatial extent and magnitude of the high runoff extremes. Regardless of regional climate

characteristics and RCPs, this behavior is expected to be enhanced under the 2.0 °C (compared with the 1.5 °C) warming scenario and increase the likelihood of flood risk (up to 10 %). More importantly, an extra 0.5 °C of global warming under 2 RCPs will amplify the change in hydroclimatic extremes on temperature, precipitation and runoff with strong robustness, especially in cold (and polar) climate zones. The results of this study clearly show the consistent changes in regional hydroclimatic extremes related to temperature and high precipitation and suggest that hydroclimatic sensitivities can differ

based on regional climate characteristics and type of extreme variables under warmer conditions over Asia.

## 1 Introduction

Due to its large population and monsoon climate, Asia is highly vulnerable to natural disasters, such as floods and droughts (International Panel on Climate Change (IPCC), 2013). The climate system in this region has changed as a result of global warming, and consequently, the frequency and intensity of natural disasters related to climate (e.g., heatwaves, heavy

precipitation, and floods) have increased (Thomas et al., 2013; IPCC, 2013; Thomas et al., 2014; Thomas et al., 2015). Moreover, further increases in atmospheric greenhouse gases (GHGs) will continue to enhance global warming and cause additional changes in the temporal and spatial patterns of both climate averages and climate extremes at the regional scale (e.g., Trenberth, 2011; Chevuturi et al., 2018). Therefore, it is essential to reliably simulate future climate changes to understand their impacts on climate extremes as well as hydrology over the Asia region.

The general approach applied to assess the impacts of climate change is to project future climate changes based on scenarios using global climate models (GCMs), downscale the resulting climate projections to a regional scale, and finally evaluate the impacts on areas of interest (e.g., in terms of the climate, water resources and climate extremes). According to this process, many previous studies have been performed on changes in climate elements and hydrologic variables under global warming

based on certain 30-year future periods defined arbitrarily by a particular time span (i.e., the near future, mid-century or a distant future period) in comparison with a 30-year reference period (e.g., Bae et al., 2011; Jung et al., 2013). These studies have reported hydroclimatic responses in future periods, but understanding how these responses are regulated is limited by the degree of current global temperature rise and by more feasible future conditions.

To avoid catastrophic consequences induced by climate change, a consensus on the warming targets of the global mean temperature was achieved in the 2015 Paris Agreement by parties to the United Nations Framework Convention on Climate Change (UNFCCC). The aim of this agreement was to keep the increase in the global mean temperature far below 2.0 ℃ above preindustrial (PI) levels and to seek targets to keep the increase within 1.5 ℃ above PI levels (UNFCCC, 2015). Therefore, recent studies have investigated both the impacts of certain warming targets (i.e., 1.5 and 2.0 ℃) on climate variables and the benefits of achieving an extra 0.5 ℃ reduction in global warming (IPCC, 2018). These studies showed robust impacts of an extra 0.5 ℃ of global warming on climate extremes (Syllar et al., 2018 and Harrington and Otto, 2018 for Europe) and on hydrologic variable-related extremes (King et al., 2017; Marx et al., 2018 for Europe; Diedhiou et al., 2018 and Nkemelang et al., 2018 for Africa; Wang et al., 2019; Kharin et al., 2018 for global). These findings reflect the necessity of understanding the impacts of global warming on climate (and hydroclimatic) extremes and the need to develop countermeasures capable of reducing the potential damage that can be induced under increases in global mean temperature.

For Asia, several studies have been conducted on the impacts of global warming on climate extremes at the continental scale. Chevuturi et al. (2018) evaluated daily natural temperature and precipitation extremes (i.e., the 99th percentile) over the Asian-Australian monsoon region and suggested that both the frequency and persistence of extremes increase in response to warming. Bhowmick et al. (2019) analyzed extreme precipitation (99th percentile) changes across South Asia for each 0.5 ℃ increase in the global mean temperature and showed changes in extreme precipitation events throughout India. Ge et al. (2019) demonstrated changes in precipitation extremes across Southeast Asia at increases between 1.5 and 2.0 ℃ and showed the responses of extreme events to increases of 0.5 ℃. On the other hand, some studies suggested that global warming would lead to extreme (e.g., temperature and precipitation) climate events at the country scale, especially in a 2.0 ℃ warming scenario compared to a 1.5 ℃ warming scenario (Li et al., 2018; Chen et al., 2018). Other studies predicted further increases in precipitation intensity and enhanced impacts on extreme precipitation in a warmer world throughout China (Zhou et al., 2018; Sui et al., 2018).

The climate conditions of Asia are influenced by a large-scale climate system (e.g., monsoon system). Nevertheless, only a few studies have assessed the extreme hydroclimatic responses to global mean temperature increases at the continental scale (i.e., Asia). Instead, the studies conducted at the country and continental scales in Asia addressed predominantly climate extremes, and thus, there are limitations to examining the hydroclimatic (e.g., precipitation and runoff) extreme responses under target global warming levels. It should be noted, however, that some studies on hydroclimatic extreme responses to global warming have been conducted at the basin scale (Zhang et al., 2019; Wen et al., 2019; Jiao and Yuan, 2019), and they suggested increases in the intensity as well as the frequency of extreme events. The hydroclimatic changes in response to global warming reflect unique regional responses because the global temperature increases impact each region differently due to changes in regional climate features. However, examining how different regional hydroclimatic extremes are caused by the impact of global warming remains challenging. To the best of our knowledge, relatively few studies have examined the impacts of global warming on extreme hydroclimatic variable-related responses considering the regional climate in Asia (Liu et al., 2019; Kim et al., 2020; Zhao et al., 2020). Therefore, the main purposes of this study are to examine the potential impacts of regional climate on hydroclimatic extremes under different global warming conditions and to investigate the regional-scale sensitivity of individual hydroclimatic variables to increases in the global mean temperature with diverse climate features. In this study, we assess the changes in climate (and hydroclimatic) extremes corresponding to the warming targets of 1.5 and 2.0 ℃ with a focus on the broad continental-scale climate zones of the Asian monsoon region (Figure 1), as delineated by Bae et al. (2013). We classify the subregions in the Asian monsoon region considering regional climate characteristics to understand

the change behaviors of climate (and hydroclimatic) extremes under global warming. To consider the reliability of future projections, we present the results based on the multimodel ensemble mean (MME) derived from five selected GCM projections, including intermodal agreement. The level of agreement among the multiple projections is used to assess the robustness (or confidence) of climate projections (Tebaldi et al., 2011; Saeed et al., 2018). This study provides scientific information for policy makers to identify regional patterns of the changes in extremes and thereby recognize the impacts of anthropogenically induced warming.

## 2 Materials and methodology

### 2.1 Study area and climate zone classification

The study area covers the Asian monsoon region with latitudes ranging from 9.75° S to 54.75° N and longitudes ranging from 60.25° E to 149.75° E, as shown in Figure 1. This region is subdivided based on regional precipitation and temperature patterns using Köppen's climate classification method (Köppen, 1936). Each subregion is categorized as a mainly tropical climate (A), arid climate (B), warm temperate climate (C), snow climate (D) or polar climate (E) according to the climate boundary conditions, which are based on threshold values of monthly temperature and precipitation (e.g., temperatures for climate zones A, C, D and E; moisture availability is required for plant growth in climate zone B). Due to its simplicity and ecologically meaningful classifications, this method has been widely used in many studies, such as assessments of the impacts of climate change on different climate characteristics (Lee et al., 2015; Fernandez et al., 2017). Table 1 shows a detailed description of the Köppen climate classification. To apply this method, we employ long-term observations (e.g., maximum temperature, minimum temperature and precipitation data) on a monthly time scale during the 30-year historical period (January 1976-December 2005). A detailed description of the observational dataset is provided in section 2.2.

### 2.2 Observational datasets

Observational meteorological datasets are required as input variables to the hydrological model on a daily time scale and for validating the performance of the GCM simulations on a monthly time scale. We select the meteorological datasets considering the availability of long-term records and their time scales. To run the hydrological simulations (1950-2005), we collect precipitation data from the Asian Precipitation Highly Resolved Observational Data Integration Toward Evaluation of Water Resources (APHRODITE) product (Yatagai et al., 2012), and the maximum and minimum temperature data and wind speed data are obtained from gridded forcing datasets provided by the University of Washington (Adam and Lettenmaier, 2003; Adam et al., 2006). To evaluate the performance of the GCM simulations, the reanalysis data for the remaining climate variables are obtained from the Coupled European Centre for Medium-Range Weather Forecasts (ECMWF) Reanalysis system-20C (CERA-20C) (Laloyaux et al., 2018) on a monthly basis due to the limited availability of data. These observational datasets, including the reanalysis data (hereafter "OBS"), are gridded at a 0.5° spatial resolution and interpolated to the same grid system as the GCMs.

### 2.3 Methodology

Figure 2 presents a flowchart of the entire procedure used in the study. To simulate the climate during both historical and future periods, climate projections forced by historical and representative concentration pathways (RCPs) 4.5 and 8.5 are selected. The five of the raw GCMs of the Coupled Model Intercomparison Project Phase 5 (CMIP5; Taylor et al., 2012) are selected by applying a unique evaluation procedure (Kim et al., 2020). Then, a reference 30-year period and two future 30-year periods of individual GCM projections are defined under warming targets of 0.48, 1.5 and 2.0 ℃ above PI levels (1861-1890) based on a time sampling method. Then, these daily forcing data (e.g., precipitation, maximum temperature, and minimum temperature) are extracted from the five selected GCM projections and then statistically bias-corrected using the

quantile mapping method. The bias-corrected GCMs are used as meteorological forcings to run the variable infiltration capacity (VIC) hydrological model. The future changes in the hydroclimatic mean and extremes corresponding to the conditions at warming targets of 1.5 and 2.0 °C are spatially analyzed according to the identified subregions based on climate

zones. We focus on the hydroclimatic extreme responses to temperature, precipitation, and runoff variations under global warming targets (i.e., 1.5 and 2.0 °C) using extreme indices. A more detailed description of each procedure is provided in section 2.4, section 2.5, and section 2.6.

## 2.4 Climate change scenarios and definitions of the periods corresponding to 1.5 °C (2.0 °C) of warming

Reliable climate change scenarios, which are derived from the selected GCMs, are important sources for estimating the impacts

of global warming on hydroclimatic (e.g., temperature, precipitation, and runoff) extremes. Here, the method for selecting GCMs suggested by Kim et al. (2020) is employed while focusing on their performance in simulating the spatial patterns of observed climate features in Asia because the regional climate is affected by physical climate system processes that occur over large spatial scales (e.g., from the planetary scale to the synoptic scale and mesoscale). For future projections, the selected GCMs are applied to the entire domain regardless of the climate zone. First, we collect 19 CMIP5 GCMs while considering

the data availability to compare each GCM's ability to represent the climatological characteristics in the study area (Figure 1). Then, for the historical climate evaluation, we use twelve relevant variables, namely, seven two-dimensional surface meteorological variables (i.e., precipitation, near-surface air mean, maximum and minimum air temperatures, outgoing longwave radiation, sea level pressure, sea surface temperature) and five three-dimensional vertical meteorological variables (i.e., air temperature, geopotential height, specific humidity, zonal wind, and meridional wind). The individual raw GCMs are

spatially disaggregated at a 0.5° horizontal resolution based on the bilinear interpolation algorithm. The GCMs are assessed in their simulation of the historical climate compared against observations (see section 2.2), namely, the climatological features of the twelve variables in the summer season (June-September) for the reference period (1976-2005). The spatial correlation coefficient (SCC) and root-mean-square error (RMSE) between the historical simulation fields derived from each GCM and the observed fields are calculated for each of the twelve relevant variables over the Asian monsoon region, as these statistics

are commonly used to examine the performance of GCMs in the simulation of observed spatial climate features (IPCC, 2013; McSweeney et al., 2015). Next, we apply the MME-based scoring rule for the selection of GCMs (Nyunt et al., 2012) to exclude low-performing GCMs and identify only the best-performing GCMs using a relative concept because the scoring rule based on the observed data does not provide the information needed to screen the GCMs. Therefore, the individual GCM statistics (i.e., the SCC and RMSE) are judged by comparison with the MME statistic. The MME statistics are considered as

criteria to score each GCM under the assumption that the MME is similar to the observed data compared with the output from only one GCM (Xu et al., 2020; Tegegne et al., 2020). The performance score of each GCM is then allocated based on the following criteria:

        1) A score of 1: the GCM has a lower RMSE and a higher SCC than the MME;

        2) A score of -1: the GCM has a higher RMSE and a lower SCC than the MME;

3) A score of 0: the GCM satisfies only one condition.

Finally, we select five GCMs, namely, bcc-csm1-1-m, CanESM2, CMCC-CMS, CNRM-CM5, and NorESM1-M, which provide the highest scores based on all the scores considering all variables, as shown in Table S1. The information of the selected GCMs is given in Table 2.

Our focus is to understand the changes in extreme hydroclimatic conditions under global warming environments of 1.5 and

2.0 °C. The timing to reach specific warming levels for individual GCMs depends on the representative concentration pathway because future projections are forced by these scenarios. The temperature response to different RCPs varies, and therefore, the increasing trend and slope of the global mean temperature differ. Here, the analysis is based on RCP4.5 and RCP8.5, which are commonly considered for realistic future projections. RCP4.5 is a stabilized emission scenario with radiative forcing of

approximately 4.5 W/m$^2$ in the year 2100, and this value is never exceeded (Thomson et al., 2011; Van Vuuren et al., 2011).

This scenario assumes that emission mitigation policies are implemented to limit emissions and radiative forcing. On the other hand, RCP8.5 is a very high emission scenario with radiative forcing of approximately 8.5 W/m$^2$ in 2100. Although the global warming process under RCP4.5, which is based on a medium-low GHG emission pathway, is relatively slow compared to higher GHG emissions (e.g., RCP8.5), many studies have suggested that the global warming climate under RCP4.5 exerts impacts on hydroclimatic phenomena (Chen et al., 2017; Donnelly et al., 2017; Kim et al., 2020). However, global warming

impacts under different RCPs on the regional changes in hydroclimatic extremes are not simple. In this regard, the results based on two RCPs (RCP4.5 and RCP8.5) can provide useful information for identifying the impacts of global warming on hydroclimatic extremes from those expected under different RCPs. This implies the need for minimum mitigation strategies as well as adaptation plans according to the global warming induced by GHG emissions, even those under the relatively low-impact RCPs (e.g., RCP4.5).

Next, for the selected five GCMs, we determine the reference period corresponding to a global mean temperature increase of 0.48 ℃ and two future periods corresponding to increases of 1.5 and 2.0 ℃ above the temperature during the PI period (1861-1890) under two RCPs using the time sampling method (James et al., 2017; Sylla et al., 2018). In this process, the individual 30-year periods and their central years (i.e., the median year of each period) are determined based on the temperature anomalies relative to the temperature of the PI period. All five GCMs reach specific warming levels in their central years and in the 30-

180 year reference and future periods under both RCP4.5 and RCP8.5 (Table 3 and Figure S1). Because the individual GCMs simulate the climate based on their own physical climate system processes, the warming phases of the GCMs are different even under the same emissions forcing. In this study, the central year of each period is the first year in which the 30-year running temperature anomaly surpasses the target temperature above the temperature of the PI period. The temperature anomalies targeted in this study are 0.48 ℃ for the reference period and 1.5 and 2 ℃ for the two future periods. To accomplish

this, the 30-year running global mean temperature is derived from the individual GCMs during the entire simulation period (1880-2100). Unlike the temperature taken from the central year of the PI period (1875), the temperature anomalies are calculated for the entire period. For the reference period, we select a warming level of 0.48 ℃, which was derived by Sylla et al. (2018) based on HadCRUT.4.6 data. The central year and 30-year periods for each GCM with global mean temperature increases of 0.48, 1.5, and 2.0 ℃ based on the two RCPs are described in Table 3. Figure S1 shows differences in the central

190 years and the global warming target periods for each RCP and GCM. The results indicate large spreads in the central year of 1.5 and 2.0 ℃ warming across all 5 GCMs under RCP4.5 relative to RCP8.5 (Zhang et al., 2019; Chen et al., 2020). The central year for the 1.5 ℃ (2.0 ℃) warming condition derived from the MME of the 5 GCMs is 2028 (2051) under RCP4.5 and 2023 (2037) under RCP8.5. Under RCP8.5, there is a shorter time lag (14 years) between the timing of 1.5 ℃ and 2.0 ℃ global warming compared to RCP4.5 (i.e., 23 years). In addition, individual global mean temperatures derived from the 5

GCMs are expected to increase above 3.0 ℃ by 2100 under RCP8.5. For the runoff simulations, each GCM with its own time period under global warming provides meteorological forcings to run the VIC hydrological model. The reference feature (denoted as REF) is derived from the MME of the selected GCMs averaged over the historical period corresponding to a warming level of 0.48 ℃. Additionally, the future 1.5 ℃ (2.0 ℃) warming feature (denoted as +1.5 ℃ and +2.0 ℃, respectively) is derived from the MME averaged over the individual 30-year periods corresponding to the central year surpassing warming levels of 1.5 ℃ (2.0 ℃).

surpassing warming levels of 1.5 ℃ (2.0 ℃).

Although we select five GCMs with relatively superior performance in the study area, there is generally inadequate accuracy in simulating the observed climate characteristics because all GCMs contain a substantial bias. Additionally, the quality of meteorological forcings (e.g., precipitation and temperatures) for the hydrological model is more important for estimating hydrological responses to climate change. Therefore, we use the quantile mapping method to reduce statistical biases in the

205 temperature and precipitation forcings on a daily basis. This method allows the whole distribution to be adjusted by matching the cumulative distribution function (CDF) of the climate model data to the CDF of the observed data, thereby improving the

mean, variance, and extreme values. This method is commonly used in many climate change studies based on climate models (MacDonald et al., 2018; Reiter et al., 2018).

## 2.5 Hydrological model

The VIC distributed hydrological model (Liang et al., 1994; 1996) is used to simulate runoff extremes in response to global warming. The VIC model simulates interactions between the land and atmosphere as well as water balances by sharing several fundamental schemes with other land surface models at the daily time step. Therefore, the VIC model is commonly coupled with a GCM not only at the continental scale but also at the global scale (Sheffield et al., 2009; Lee et al., 2015). We establish the VIC model at a spatial resolution of 0.5° (approximately 50 km) considering the study domain and run the model on a daily

basis, as was suggested in Bae et al. (2015).

In addition, we collect geophysical datasets that are required for the VIC model, that is, digital elevation model (DEM) data from the United States Geological Survey (USGS), soil data from the Food and Agriculture Organization (FAO, 1998), and land use data from the University of Maryland (Hansen et al., 2000). The collected datasets are converted to a 0.5° grid resolution to conform to the spatial resolution of the VIC model.

Because runoff simulation results depend on the model parameters, it is important to calibrate and verify the hydrological model parameters to obtain a reliable runoff simulation (Bae et al., 2011). Some model parameters are estimated based on geophysical datasets and river networks for gauged basins, but the remaining parameters for ungauged basins are estimated indirectly by using the hydrological regionalization method (Parajka et al., 2013; Bae et al., 2015; Beck et al., 2016). We apply the hydrological regionalization method by transferring parameters obtained from gauged regions to ungauged regions based

on the assumption that two basins with analogous climate features (e.g., based on the climate zone classification) exhibit similar hydrological responses. For runoff simulations at the global scale, Nijssen et al. (2001) obtained the parameters for an ungauged basin from the estimated parameters of a gauged basin with the same temperature and precipitation features. Xie et al. (2007) and Bae et al. (2013) employed the same approach leveraging climatological similarity according to Köppen's climate classification method and suggested the applicability of this method over China and Asia, respectively. In this study,

both gauged basins and ungauged basins are divided into one of the climate zones to apply the hydrological regionalization method. We examine the optimal parameters for individual climate zones that effectively simulate runoff based on the estimated parameter sets obtained from all gauged basins within each climate zone. The optimal parameters of each climate zone are then transferred to the ungauged basins belonging to the same climate zone. In our previous study, the regionalization results were verified by assuming that some gauged basins are considered ungauged basins (Bae et al., 2013), and the results

support the adaptability and applicability of the VIC model to simulate runoff throughout our study area.

The model parameters are estimated based on gridded runoff; therefore, we assume that the time delay described by the channel routing scheme is not significant considering the horizontal grid resolution. To evaluate the reliability of the runoff results, the simulated mean and extreme runoff (i.e., monthly maximum runoff) values are validated by comparison with measured data. In this study, the simulated runoff is driven by observational meteorological forcings for the historical period (1950-2005) to

240 compare the historical runoff records obtained from the Global Runoff Data Centre (GRDC). Some parameter validation results for the VIC model in 20 river basins (Figure S2) considering the data availability of measurement records are suggested in Table S2, Figure S3 and Figure S4, and additional results can be found in a previous study (Bae et al., 2013). The simulated monthly mean runoff obtained from the VIC model using observational meteorological input data shows a high temporal correlation with the observed pattern for 6 basins (see Figure S3), and the range of correlation coefficients over the 20 basins

is 0.58~0.97 (see Table S2). To evaluate the accuracy of the VIC model, we also consider other quantitative statistics, such as the model efficiency (ME), root-mean-square error (RMSE), and volume error (VE), as shown in Table S2. In general, simulated runoff qualitatively and quantitatively simulates the measured runoff. Figure S4 presents the scatter plot and box-whisker diagram of measured and simulated monthly maximum runoff in the 20 basins. The assumptions used in parameter

estimation and runoff analysis at the continental scale may impact the uncertainty in simulating monthly maximum runoff (see Figure S4a and Figure S4b), especially in capturing extreme runoff periods. Because it is inherently more difficult to simulate long-term mean runoff extremes using a hydrologic model, uncertainty exists between the simulated and measured extreme runoff data. Although simulated monthly maximum runoff (denoted as SIM) tends to underestimate the measured values (denoted as OBS), SIM commonly reproduces the OBS in terms of the interquartile range (see Figure S4b) and the biases compared to the variation range of OBS (see Figure S4c). The results can aid in understanding runoff features when observational data are not available, even though the results are limited when simulating realistic runoff.

### 2.6 Extreme indices

Fixed-threshold indices are needed as extreme indices for the purpose of comparing changes in hydroclimatic extremes among different climate regions under target global warming conditions. We selected four extreme temperature indices, six extreme precipitation indices, and three extreme runoff indices for extreme climate and runoff analyses (Table 4). The extreme indices used in this study are widely accepted for extreme analyses (Dosio and Fischer, 2018). For the changes in temperature extremes, the numbers of tropical days (TR), frost days (FD), warm nights (TN90P), and cold nights (TN10P) are calculated using daily minimum temperature data during the reference period and two future periods for each selected GCM, as shown in Table 3. The numbers of summer days (SU), ice days (ID), warm days (TX90P), and cold days (TX10P) are calculated by daily maximum temperature data. The extreme indices associated with daily precipitation are very wet day precipitation (P95), extreme wet day precipitation (P99), annual maximum precipitation (PX1D), and maximum precipitation over 2, 3, and 5 consecutive days (PX2D, PX3D, and PX5D, respectively). Finally, the variables associated with extreme runoff, as suggested by Nandintsetseg et al. (2007), are the minimum consecutive 7-day and 30-day runoff (DWF07 and DWF30, respectively) and the annual maximum runoff (MDF). Table 4 provides detailed information on the extreme indices used in this study.

### 3 Results

#### 3.1 Classification of climate zones and validation of the reference simulation

The climate zones over the Asian monsoon region in this study are classified based on long-term (30-year; 1976-2005) observation datasets (i.e., precipitation from APHRODITE; minimum and maximum temperatures from the University of Washington). Figure 1 shows the classified climate zones obtained by applying Köppen's climate classification method. The study domain (i.e., the Asian monsoon region) is divided into twelve climate zones. The tropical climate zone (A) encompasses the low latitudes of Indonesia, Malaysia, the Philippines, and Thailand (Aw), the northwestern parts of India and Myanmar (Am; located between Af and Aw), and the northern parts of Indonesia, India, Vietnam, Thailand, and Myanmar (Aw; located between 9° N and 25° N). The arid climate zone (B) includes northwestern China and some parts of Mongolia, India, Pakistan, and Afghanistan (BS), as well as northern China, southern Mongolia, Pakistan, and Kazakhstan (BW). The warm temperature climate zone (C) appears in central and northern India and some parts of Afghanistan (Cs); the southern and eastern parts of China, the northern parts of India, Vietnam, Thailand, and Myanmar, and the southern part of South Korea (Cw); and most of southeastern China, the coastal region of South Korea, and the southern part of Japan (Cf). The cold climate zone (D) spreads over the northern part of Afghanistan (Ds), northeastern China, and most of the inland region at high latitudes (Dw and Df) above 38° N. The tundra climate zone (ET) appears on the Tibetan Plateau and the Himalayas. The largest number of grid points in the Asian monsoon region are in zone D, followed by zones B, C, A, and E, and the ratio for each region is listed in Table 1.

Prior to the assessment of the influence of global warming on the hydroclimatic extremes in the Asian monsoon region based on the GCM projections, the bias-corrected GCMs are validated to determine whether GCM simulations can adequately

represent the historical climatological characteristics noted in the observed changes. Precipitation data are obtained from the MME of multiple GCMs and APHRODITE at the grid points in the study area (see Figure 1) for a long-term period (1950-2005). Hereinafter, the results based on the MME of the selected five GCMs and from APHRODITE are referred to as MME and OBS, respectively.

Figure 3 depicts the spatial distributions of the climatological annual mean precipitation (hereafter referred to as PANN) and the climatological annual maximum precipitation (hereafter referred to as PX1D) of the OBS and MME (1976-2005). The percentage bias (hereafter referred to as BIAS) between the OBS and MME is calculated to examine the quantitative error in the MME. The MME properly captures both the spatial pattern and the magnitude of PANN and PX1D (Figure 3a, b). The relatively large magnitude of bias in PANN (PX1D) is shown in the region with low PANN (PX1D). Despite the similarity in the PX1D values between the OBS and MME, the MME shows a tendency to slightly overestimate the OBS PX1D for Southeast Asia and Southeast China (within a PX1D range of 45-90 mm/day), as presented in Figure 3b. Although there is a deficiency between the OBS and MME precipitation values, the MME, which is derived from the bias-corrected GCMs, reflects the OBS characteristics of both PANN and PX1D. The validation results of the MME compared with the OBS for the minimum (and maximum) temperature are illustrated in Figure S5. The MME outputs of the minimum and maximum temperatures are very similar to the OBS temperature patterns. In addition, the simulated runoff based on the MME and OBS are compared due to the lack of measured runoff data (Figure S6). The MME results show reasonable historical simulations with implications for the reliability of the climatological and hydrological responses to the climate forcing derived from the MME.

## 3.2 Future projections of temperature extremes under 1.5 and 2.0 ℃ of warming

We examine the future changes in temperature extracted from the MME according to global warming. We calculate the changes in the extreme temperature indices under two global warming scenarios on the basis of a relative concept, that is, the difference between the reference period (REF) and each target condition (+1.5 ℃ and +2.0 ℃). We identify the regions with absolute intermodel agreement in the change signals, which shows a high degree of consistency among the results from the different GCMs.

Figure 4 shows the relative changes in the cold extreme indices (FD, ID) and warm extreme indices (SU, TR), which are derived from the MME between the warming conditions (i.e., 1.5 and 2.0 ℃) under RCP4.5 and REF over Asia. In total, consistent patterns are observed for the temperature changes with decreasing change patterns for the cold extreme indices (FD, ID) and increasing change patterns for the warm extreme indices (SU, TR), with 5 out of 5 model agreements under both 1.5 and 2.0 ℃ warming conditions. The change patterns in the temperature extreme indices over Asia are amplified under 2.0 ℃ of warming compared with those under 1.5 ℃ of warming, as was suggested in a previous study (e.g., Chevuturi et al., 2018; Sui et al., 2018). In particular, the cold extreme indices (FD, ID) exhibit large decreases in the mid-latitude region (above 25° N) compared to the low-latitude region (below 25° N). Moreover, tropical nights (TR) show large increases in the low-latitude region (below 25° N). An increase in summer days (SU) is dominant in most regions except for the low-latitude region (below 25° N). However, some indices show no changes in some areas because the changes in the extreme temperature indices are estimated based on fixed-threshold criteria (Dong et al., 2018). For example, the low-latitude regions (below 25° N; A zones) with high maximum and minimum temperatures do not present changes in either the cold extreme indices (Figure 4a, b) or the warm extreme indices (Figure 4d). On the other hand, the ET zone and high-latitude region (above 40° N) with low temperatures do not show changes in SU or TR, respectively (Figure 4c, d) because in this region, even though the global mean temperature is increased by 1.5 ℃ (2.0 ℃) compared to PI levels, the daily temperatures on some days are not sufficiently large to reach the criterion of warm extreme indices (i.e., TN exceeding 20 ℃). These features (e.g., change patterns and spatial distributions) are shown in the results under RCP8.5 (related figure not suggested here).

Figure 5 shows the area-averaged changes in the cold and warm extreme indices derived from the results under RCP4.5 shown in Figure 4 (and under RCP8.5); these area-averaged values are derived from the values averaged over all grid points included in each classified climate zone. The change in FD over Asia represents the largest decrease of approximately -10.0 days at 1.5 ℃ of warming and -14.1 days at 2.0 ℃ of warming under the two RCPs. The change in ID also decreases by approximately -6.4 days at 1.5 ℃ of warming and -9.0 days at 2.0 ℃ of warming under the two RCPs. A large reduction in both FD and ID

is detected in the cold climate zones (Ds, Dw, and Df) and polar climate zones (ET) with lower temperature records than the other climate zones. In contrast, the change in TR over Asia represents the largest increase of approximately 13.6 days (15.0 days) at 1.5 ℃ of warming and 20.6 days at 2.0 ℃ of warming under the two RCPs. Similarly, the change in SU is an increase of approximately 11.2 days at 1.5 ℃ of warming and 15.7 days at 2.0 ℃ of warming under the two RCPs. While the difference in the value of the results from the RCPs is the largest (i.e., approximately 1.4 days) in TR, it is similar in the other temperature

extremes (i.e., FD, ID and SU). The large magnitudes of change in TR and in both TR and SU are found in the tropical zones (Af, Am, and Aw) and in the warm temperature climate zones (Cs, Cw, and Cf), respectively. In general, larger changes in the cold and warm extreme indices under 1.5 ℃ warming compared to the REF period are found under RCP8.5 relative to RCP4.5. Relatively small differences in these changes are found between RCP4.5 and RCP8.5 for the 2.0 ℃ warming condition. Understanding the change behavior of the daily temperature is necessary for detecting a linkage to extreme temperature events.

We calculate the relative changes in the frequency of both daily maximum and daily minimum temperatures between individual warming conditions (1.5 and 2.0 ℃) and the REF period based on the initial percentile range (e.g., 10th, 50th, and 90th percentile values in the REF period). Figure 6 presents the distributions of the low-percentile and high-percentile temperatures relative to the changes in whole temperature events under 1.5 and 2.0 ℃ of warming over Asia, with 5 out of 5 model agreements. In all climate zones, increased high-percentile temperatures (above the 50th percentile) frequently occur at the

expense of reduced low-percentile temperatures (below the 50th percentile) under a warmer climate. In addition, this trend is clear in the exceedance of extremes (e.g., below the 10th percentile or above the 90th percentile). Warm days (TX90P) over Asia are projected to increase by 27.4 % under 2.0 ℃ of warming and by 18.7 % under 1.5 ℃ of warming for the two RCPs. Moreover, warm nights (TN90P) are projected to increase by 33.0 % under 2.0 ℃ of warming and by 23.6 % under 1.5 ℃ of warming under the two RCPs. The rate of warm days (TX90P) increase and warm nights (TN90P) increase are higher under

RCP8.5 compared to RCP4.5. Conversely, cold days (TX10P) are projected to decrease by -7.4 % above PI levels on average in Asia at 2.0 ℃ of warming and by -6.1 % at 1.5 ℃ of warming under the two RCPs. Cold nights (TN10P) are projected to decrease by -8.3 % under 2.0 ℃ of warming and by -7.1 % under 1.5 ℃ of warming under the two RCPs. The rate of cold days (TX10P) decrease and cold nights (TN10P) decrease are slightly steeper under RCP8.5 than under RCP4.5. A large disparity between RCP4.5 and RCP8.5 is found in the change patterns of TX90P above the 50th percentile compared to TN90P.

Overall, these change features in TN are more intense than those in TX (Figure 6a, c), which agrees with previous findings (IPCC, 2018).

However, changes in temperature under global warming are associated with latitude rather than regional climate characteristics (Dong et al., 2018). The TX90P (TN90P) change patterns derived from the MME are related to the area-averaged latitude in each climate zone (Figure 6b, d). The negative relationship between TX90P (TN90P) and the area-averaged latitude indicates

that marked increases in the extreme hottest temperatures (e.g., exceeding the 90th percentile of daily maximum and daily minimum temperatures; TX90P and TN90) occur more frequently in low-latitude regions. Among the 12 climate zones, the largest changes in both TX and TN are observed in tropical climate zones (Af, Am, and Aw). These results imply that tropical climate regions (which exhibit the lowest interannual temperature variability) are very sensitive to warm temperatures, as was demonstrated in the IPCC (2018). This robust behavior is more prevalent in TN90P because its sensitivity to an increasing

global temperature is higher than that of TX90P. Overall, global warming above PI levels affects strong changes in the distributions of the maximum and minimum temperatures (TX and TN) on a daily time scale, and the projected changes trend toward an enhancement at high-percentile temperatures compared to the REF period regardless of the climate characteristics,

especially under RCP8.5 compared to RCP4.5, which may in turn lead to increased risks of heatwaves as well as temperature-based seasonal cycle changes.

## 3.3 Future projections of precipitation extremes under 1.5 and 2.0 ℃ of warming

Anthropogenic forcings have been attributed to the intensification of regional precipitation extremes (e.g., O'Gorman, 2015; Weber et al., 2018; Guo et al., 2016). Here, we examine future changes in precipitation from the MME of the five GCMs under two global warming scenarios (i.e., 1.5 and 2.0 ℃). The regions with 100 % and 80 % intermodel agreement on the change patterns are identified and employed for the analyses in sections 3.3 and 3.4 to provide robust future change patterns.

Under the two selected RCPs (i.e., RCP4.5 and RCP8.5), Figure 7 displays the relative changes in the extreme precipitation indices (very wet day precipitation and extreme wet day precipitation; P95 and P99, respectively) with regard to its amount, frequency and intensity under 1.5 and 2.0 ℃ of warming in comparison to the REF period, indicating that global warming tends to intensify the amount, frequency and intensity of extreme precipitation over Asia. Overall, consistent increases in both very wet day precipitation (P95) and extreme wet day precipitation (P99) under 1.5 and 2.0 ℃ conditions are detected in most of the climate zones. In particular, the increasing change patterns of both P95 and P99 at 2.0 ℃ of warming are stronger than those at 1.5 ℃ of warming (Figure 7a, b, c). In most regions, the changes in P99 are larger and more robust with regard to the total amount, frequency and intensity than those in P95. The largest difference between P95 and P99 is the alteration in the intensity, while the magnitudes of change are the lowest in terms of the intensities of both P95 and P99 rather than the total amount or frequency, the robustness of the intensity change is the highest. Compared to changes in extreme temperature indices, small differences between the two selected RCPs are found for P95 and P99 in terms of the total amount, frequency, and intensity.

Figure 8a and Figure S7a present the spatial distributions of the change in the annual maximum precipitation (PX1D) under 1.5 and 2.0 ℃ of warming based on RCP4.5 and RCP8.5, respectively, in comparison with that under the REF period; a consistent increasing pattern is found for PX1D (except for several grids that showed reduced changes under both 1.5 and 2.0 ℃ conditions). As the globe warms under RCP4.5, the intensity of extreme precipitation consistently increases in most regions of Asia (93.1 % of the whole domain at 1.5 ℃ of warming and 96.8 % of the whole domain at 2.0 ℃ of warming). As shown in Figure 8a, the increasing patterns over the study area become more apparent and robust (with 4 out of 5 model agreements) under 2.0 ℃ than under 1.5 ℃ of warming. This finding implies an intensification of extreme precipitation. Most of the grids exhibiting an increasing pattern in PX1D over Asia are likely to show increases in 2, 3, and 5 consecutive days of maximum precipitation (PX2D, PX3D, and PX5D) under both 1.5 and 2.0 ℃ of warming in comparison to the REF period. The spatial distributions of the change patterns in PX2D, PX3D, and PX5D are similar to those of PX1D under both 1.5 and 2.0 ℃ (Figure 8b and Figure S7b). Under both RCPs, the pattern correlation coefficient (PCC) values between PX1D and PX2D, PX3D, and PX5D are 0.89, 0.83, and 0.73, respectively. In addition, the PCC differences between 1.5 and 2.0 ℃ of warming are not robust in all cases. As shown by these PCC results, the change pattern of PX1D is highly correlated with the change patterns of the other indices (i.e., PX2D, PX3D, and PX5D) in terms of the spatial distribution. These results describe the intensification of extreme precipitation with similar spatial behaviors under warmer climate conditions.

Figure 9 presents the area-averaged changes in annual mean precipitation (PANN) and PX1D compared to the REF period under 1.5 and 2.0 ℃ warming conditions based on RCP4.5 (RCP8.5). The changes in PX1D are greater than the changes in PANN in most climate zones except Bs and Bw (shown in Figure 8a and Figure S7a) under both RCP4.5 and RCP8.5. An increase in PANN under global warming based on the two RCPs compared with the REF period ranges from 0.1 % to 10.7 % at 1.5 ℃ of warming and from 11.7 % to 11.9 % at 2.0 ℃ of warming. Similarly, under the two RCPs, PX1D is projected to significantly increase from 5.7 % to 11.2 % under 1.5 ℃ of warming and from 8.0 % to 15.2 % under 2.0 ℃ of warming. Namely, warming of 2.0 ℃ results in higher precipitation than warming of 1.5 ℃ in terms of both the PANN and PX1D irrespective of RCP scenarios. Under warmer climate environments, PX1D is expected to increase in all climate zones with a

high level of robustness compared to PANN. Hence, global warming will lead to adverse influences on the risk of flooding over Asia due to increased high-intensity precipitation events, especially under 2.0 ℃ of warming.

## 3.4 Future projections of runoff extremes under 1.5 and 2.0 ℃ of warming

In this section, we examine the future changes in runoff based on the VIC simulations, which are fed with the five individual GCMs. Figure 10 (Figure S8) indicates the spatial distributions of the changes in the high and low runoff extreme indices over
Asia based on the 1.5 and 2.0 ℃ warming scenarios under RCP4.5 (RCP8.5). The consistent patterns (with 4 out of 5 model agreements) reflect an increase in the annual maximum runoff (MDF) across most regions (except for several grids under the 2.0 ℃ warming condition). This result implies intensified extremely high runoff, which may increase the risk of flooding. In contrast, the low runoff indices (minimum consecutive 7-day and 30-day runoff; DWF07 and DWF30, respectively) exhibit different change patterns in different regions under 1.5 and 2.0 ℃ warming conditions throughout Asia. As warming intensifies,
increases in both DWF07 and DWF30 become more dominant than decreases in both DWF07 and DWF30 over Asia. However, as indicated by the regions highlighted green and yellow in Figure 10 (Figure S8), some regions (the Cf and Bw zones) show consistent decreasing patterns for both the 1.5 and 2.0 ℃ warming conditions. Under RCP8.5, the decreasing signal of DWF07 (DWF30) is additionally shown in Af zones (e.g., highlighted purple region in Figure S8). As the temperature increases, these regions are likely to be susceptible to changes in DWF07 and DWF30. Because precipitation patterns are converted into runoff
features (Kim et al., 2020), changes in the spatial distributions and increasing (or decreasing) patterns of the runoff indices are highly similar to the changes in the precipitation indices, especially the change patterns of annual maximum precipitation (PX1D) and high runoff indices (MDF).

Figure 11 presents the area-averaged annual mean runoff (hereafter referred to as RANN) and high runoff indices (MDF) compared to the REF period under the 1.5 and 2.0 ℃ warming conditions based on RCP4.5 and RCP8.5. Changes in the annual
mean runoff (RANN) increase in all climate zones under global warming compared with the REF period, and the change pattern of MDF also increases in most of the climate zones except the Ds zone, which shows a large variation among the five GCMs. The magnitude of the change in MDF over Asia is projected to be greater than that in the REF period, especially under the 2.0 ℃ warming condition compared to the 1.5 ℃ warming condition, in the majority of the climate zones (i.e., all A, C, and E zones, the Bw zone, and the Ds zone). Warming of 2.0 ℃ causes a sharp increase in runoff in terms of both RANN and
MDF compared with warming of 1.5 ℃, which implies the intensification of runoff with global warming. As with changes in precipitation, an increase in MDF over all climate zones shows a considerable degree of robustness compared to RANN at both 1.5 and 2.0 ℃ of warming. Warming over Asia will aggravate the management of water resources due to these challenging situations, for example, an increase in MDF and a large spatial disparity of changes between DWF07 and DWF30.

## 4 Discussion and conclusions

As suggested by the IPCC, anthropogenic influences have likely affected the global climate system, and such effects increase the likelihood of intensified extreme climate events (e.g., heatwaves, precipitation, flooding, and droughts) worldwide (IPCC, 2013; 2018). An extreme climate event is a phenomenon that occurs at a level above (or below) a threshold defined by a normal range within a given region for each variable. In addition, the extrema are closely related to the climate features of certain regions. Therefore, to minimize the damage from climate disasters under global temperature increases of 1.5 and 2.0 ℃ above
PI levels, it is important to analyze regional changes in both long-term climate patterns and climate extremes (Kharin et al., 2018; Dong et al., 2018).

Table 5 presents the relative changes (%) in the average and extreme hydroclimatic indices under a further 0.5 ℃ increase in temperature from the difference in the global mean temperature in each climate zone between the 1.5 and 2.0 ℃ warming scenarios for RCPs (i.e., RCP4.5 for Table 5a and RCP8.5 for Table 5b). Based on both RCP4.5 and RCP8.5, all the changes

in the individual hydroclimatic extremes except for the runoff indices (MDF, DWF07, and DWF30) exhibit similar change signals in all the climate zones under an extra 0.5 ℃ of warming. However, the influence of an additional increase of 0.5 ℃ of warming on the hydroclimatic extremes shows diverse change patterns and magnitudes with regard to different regions and types of extreme climate indices. Temperature extremes present the same change signals (e.g., an increase or a decrease) with a high degree of robustness over all climate zones. As the globe warms, changes in the warm extreme indices (e.g., SU, TR, TX90P, and TN90P) exhibit an increasing trend over most climate zones, except for SU and TR in the Af and ET zones. On the other hand, the cold extreme indices (e.g., FD, ID, TX10P, and TN10P) have a distinct tendency to increase across Asia. Large increases in the extreme warm indices are observed in arid climates (BS and BW) and cold climates (Ds, Dw and Df), whereas they are projected to decrease considerably in warm temperate climates (Cs, Cw and Cf). Furthermore, the changes in the extreme precipitation indices (e.g., P95, P99, PX1D, PX2D, PX3D, and PX5D) exhibit large increasing patterns compared to PANN with an extra 0.5 ℃ of warming. The change in MDF is similar to that in PX1D because the change patterns of precipitation influence those of runoff. These results represent an increase in the risks of runoff and flooding in most climate zones over Asia. The changes in the Cs and Ds zones with dry summer features show somewhat greater variability than the other climate zones in terms of both the average and the extreme precipitation (and runoff) indices under a climate environment characterized by an extra 0.5 ℃ of warming. In general, in comparison with those of MDF, the change patterns of the low runoff indices (DWF07 and DWF30) show relatively less robust patterns, especially in terms of the lower magnitude of change and decreasing change patterns (e.g., MacDonald et al., 2018). Although the future projections of low runoff contain levels of uncertainty due to variations among the individual GCMs, the Cf and Bw zones are likely to face challenges in coping with low runoff under global warming (Figure 10 for RCP4.5 and Figure S8 for RCP8.5).

However, zones D and E are highly susceptible to an extra 0.5 ℃ of warming. These regions show robust changes in temperature extremes, high-precipitation extremes and high-runoff extremes, as depicted in Table 5. Under RCP4.5 (RCP8.5), the area-averaged cold extremes in this region are expected to decrease by -4.0 % (-2.8 %) in FD and -6.8 % (-5.2 %) in ID, while the area-averaged warm extremes are projected to vastly increase by 57.2 % (50.8 %) in SU and 80.8 % (68.3 %) in TR. Similarly, the high-precipitation extremes are projected to increase by approximately 3.3~3.6 % (1.1~1.9 %) for PX1D, PX2D, PX3D, and PX5D and approximately 10.5 % (5.6 %) and 18.7 % (9.8 %) for P95 and P99, respectively. Consequently, the high-runoff extremes (i.e., MDF) are expected to increase by 3.4 % (0.3 %) under RCP4.5 (RCP8.5), which is likely to result in a risk of more intensified flooding. In contrast, the changes in the low-runoff extremes (DWF07 and DWF30) show low robust change signals in these regions as a result of small changes under a further 0.5 ℃ of global warming and substantial uncertainty in the GCM projections; this finding agrees with previous results (e.g., Chen et al., 2017; Donnelly et al., 2017; Marx et al., 2018). However, the change behavior in the hydroclimatic extremes (except for the low-runoff extremes) tends to be amplified at 2.0 ℃ of warming compared with 1.5 ℃ of warming regardless of the RCP. Although substantial changes in the characteristics of the various extreme indices are found under RCP8.5, the small differences in these change patterns between the two selected RCPs are evidenced by the large changes under the 1.5 ℃ warming condition in comparison to RCP4.5. More importantly, under RCP8.5, global warming is likely to occur faster, and the degree of warming is much higher (e.g., above 3.0 ℃ of global warming) compared to RCP4.5. Our results imply the necessity for mitigation to alleviate the negative impacts of anthropogenic warming and to reduce the increased risk of hydroclimatic extremes under a far warmer climate.

Next, we focus on the changes in hydroclimatic extremes across diverse climate zones over Asia in response to warming scenarios of 1.5 and 2.0 ℃ under two emission forcings (RCP4.5 and RCP8.5) above the PI level. Five CMIP5 GCMs are selected considering their performance in the historical simulations. The central years and 30-year periods reaching warming targets of 1.5 and 2.0 ℃ are identified based on the individual GCMs. After removing systematic biases, five GCMs are used as meteorological forcings for the VIC distributed hydrological model, and the simulated surface runoff is converted into area-averaged runoff according to each climate zone. Future changes in various extreme indices (e.g., temperature, precipitation,

runoff-related indices) are calculated by applying the relative concept to the differences between the individual warming conditions (1.5 and 2.0 ℃) and the REF period. Our focus is to estimate and compare the change patterns of the extreme temperature, precipitation, and runoff indices among the various climate zones under 1.5 and 2.0 ℃ of global warming.

In all climate zones, an extra 0.5 ℃ of global warming has a considerable influence on the changes in hydroclimatic extremes. The changes in temperature indices show the strongest robustness over Asia (with 5 out 5 model agreements) and project a greater increase in high-percentile maximum and minimum temperatures. Although there is great uncertainty in the precipitation and runoff projections, the high-precipitation and high-runoff extremes show increasing patterns with a high level of robustness.

This finding supports the concept that global warming leads to an intensified hydrological response, such as an increase in high-precipitation extremes (e.g., Trenberth, 1999; Giorgi et al., 2014; Im et al., 2017; Kim et al., 2020). Consequently, consistent with the change patterns of precipitation extremes, high-runoff extremes under warmer conditions are likely to increase the risks of water-related disasters in most climate zones of Asia. Our findings are generally consistent with previous studies that have suggested likely increases in high-runoff extremes under warmer climate conditions above PI levels (e.g., MacDonald et al., 2018; Jacob et al., 2018; Paltan et al., 2018; Kim et al., 2020). Finally, although these behaviors are taken from a limited number of GCMs, our CMIP5 MME-derived findings reveal accelerated extremes compared to the long-term mean. Since hydroclimatic sensitivity differs based on regional climate characteristics, understanding the change behaviors of hydroclimatic extremes is clearly required at the regional scale. As shown in Table 5, the unique regional responses (with high significance measured by the intermodal agreement level) of an extra 0.5 ℃ of global warming reveal the need for different adaptive measures to expected hydroclimatic extremes. Although the vulnerability of temperature extremes will be increased in all climate zones over Asia, the frequencies of summer days and tropical nights are increased by 10 % and 20 %, respectively, in cold climate regions (D zones) under extra global warming. This temperature-related risk is likely to increase the adverse effects on human health, such as heat-related illnesses. Regarding precipitation extremes, adaptation for intensified heavy rainfall in terms of both frequency and intensity will be needed in most climate zones except for some climate regions with dry summer features (e.g., BW, Cs, and Ds). Changes in heavy rainfall amplify the risks associated with flood extremes and consequently flood damage (e.g., loss of life and economic losses). The daily maximum runoff, which is related to flood hazards, will be increased by 4~8 % in zones Cw, Cf, Dw, and ET. Therefore, both structural (e.g., flood-adaptive design for hydraulic structures) and nonstructural measures (e.g., flood forecasts and measurements) are needed for flood risk management in these regions. Although the potential impacts of low-runoff extremes (e.g., minimum consecutive 7-day and 30-day runoff) show low significance in all classified climate zones under extra global warming, the low-runoff extremes are amplified by more than 10 % at 2.0 ℃ of global warming compared to 1.5 ℃ of global warming in the western parts of India and the high latitudes (above 40° N), thus increasing the risk of water supply issues for drinking and irrigation as well as drought conditions. As the global temperature increases, regional climate change impacts hydroclimatic conditions and related aspects (e.g., human health, water supply, water-related disasters, hydraulic structures). These results suggest positive benefits of 0.5 ℃ less warming in terms of hydroclimate extremes and the necessity of adaptive regional planning.

**Code and data availability**

The VIC model code is publicly available at https://vic.readthedocs.io/en/master/SourceCode/Code/. The USGS DEM data (i.e., GTOPO30) are available at https://earthexplorer.usgs.gov/. The meteorological forcing data used in this study can be found in the APHRODITE (http://www.chikyu.ac.jp/precip/english/products.html) and global datasets of the VIC model (https://vic.readthedocs.io/en/master/Datasets/Datasets/) and at the ECMWF website (CERA-20C;

https://www.ecmwf.int/en/forecasts/datasets/reanalysis-datasets/cera-20c). More information on the data can be obtained by emailing the corresponding author or first author.

**Author contributions**

JBK designed and carried out the study. DHB supervised the project. All the authors contributed to the writing and interpretation of the results.

**Competing interests**

The authors declare that they have no conflicts of interest.

**Acknowledgments**

This work was supported by Korea Environment Industry & Technology Institute(KEITI) through Water Management Research Program, funded by Korea Ministry of Environment(MOE) (130747) and supported by Ministry of Science and ICT. The authors send thanks to Rohini Kumar, Shah Harsh and Anonymous Referee for valuable comments and suggestions. We acknowledge the World Climate Research Programme's Working Group on Coupled Modeling, which is responsible for CMIP, and we thank the climate modeling groups (listed in Table 2 of this paper) for producing and making available their model output. For CMIP the U.S. Department of Energy's Program for Climate Model Diagnosis and Intercomparison provided coordinating support and led the development of software infrastructure in partnership with the Global Organization for Earth System 350 Science Portals.

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

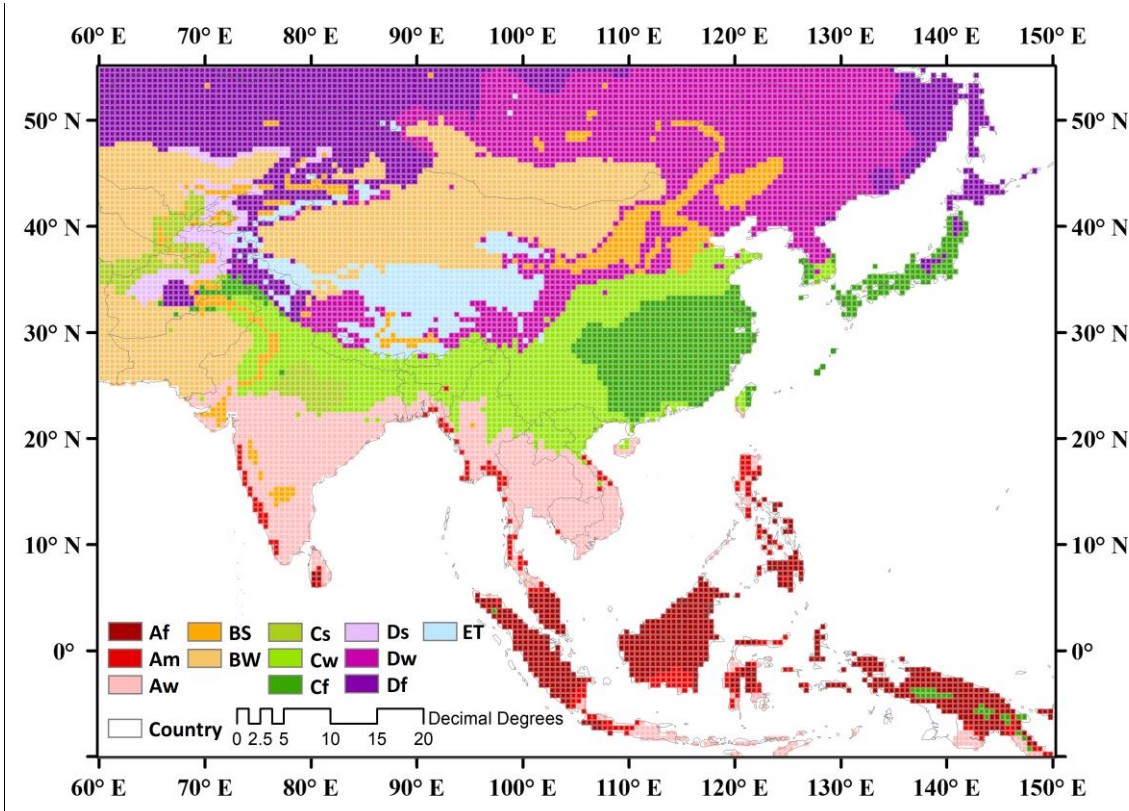

**Figure 1: Study domain, climate zone classifications, and the 10,977 grid points of the VIC model used for the analyses in this study. National boundaries are delineated by black lines. The 12 climate zones are based on the Köppen climate zone method and are denoted by individual colors.**

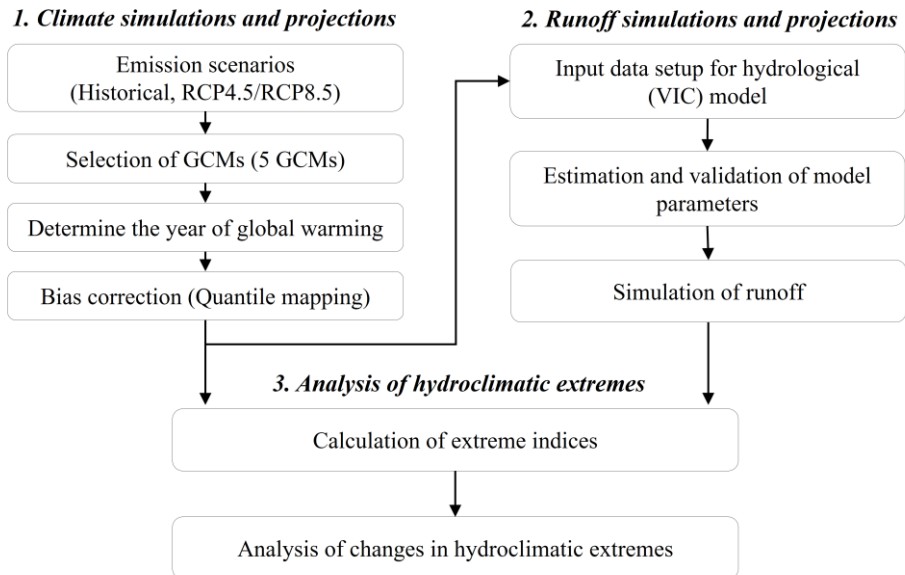

**Figure 2: Flowchart of the entire procedure used in this study.**

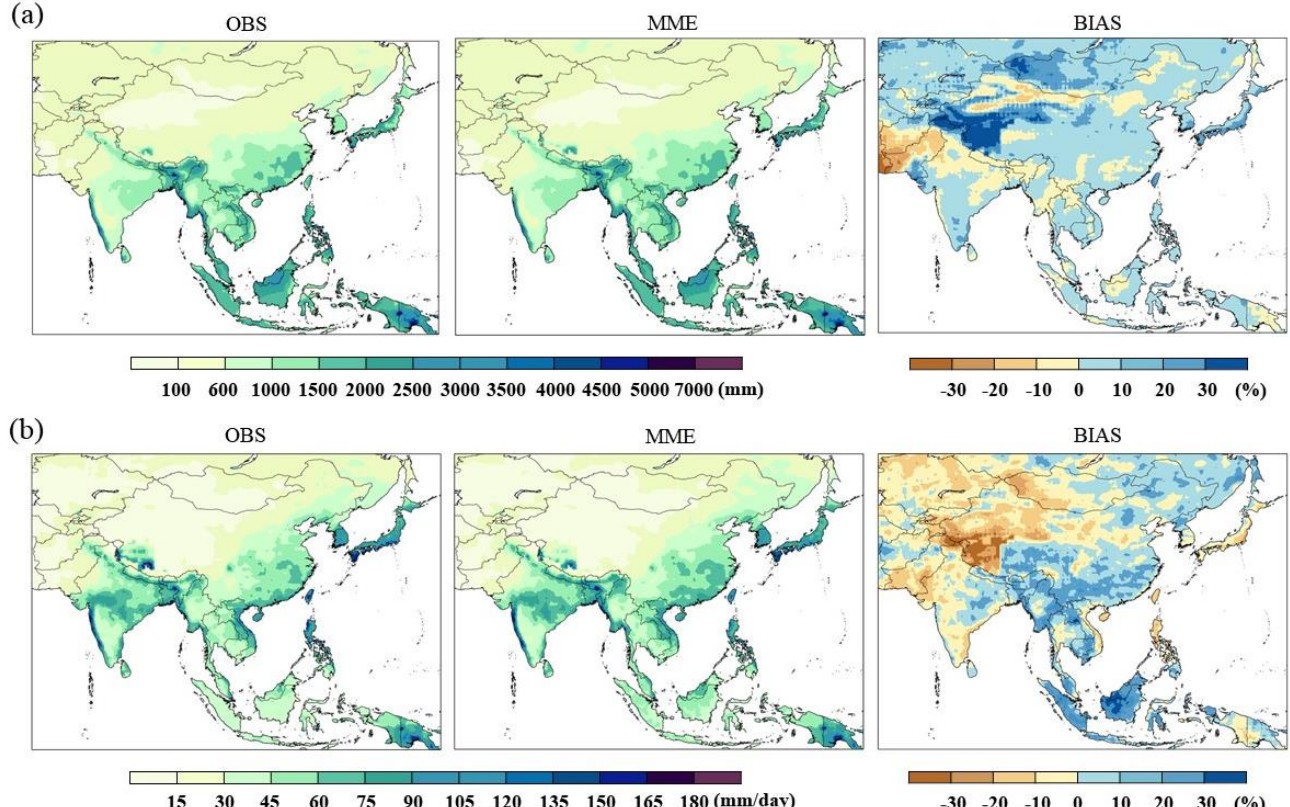

**Figure 3: Spatial distributions of the (a) annual mean precipitation (PANN) and (b) annual maximum precipitation (PX1D) for the historical period (1976-2005) in the Asian monsoon region derived from observations (OBS) and the MME of bias-corrected outputs from the five GCMs. BIAS (i.e., the 3rd column in each row) represents the percentage bias in PANN (PX1D) between OBS and MME.**

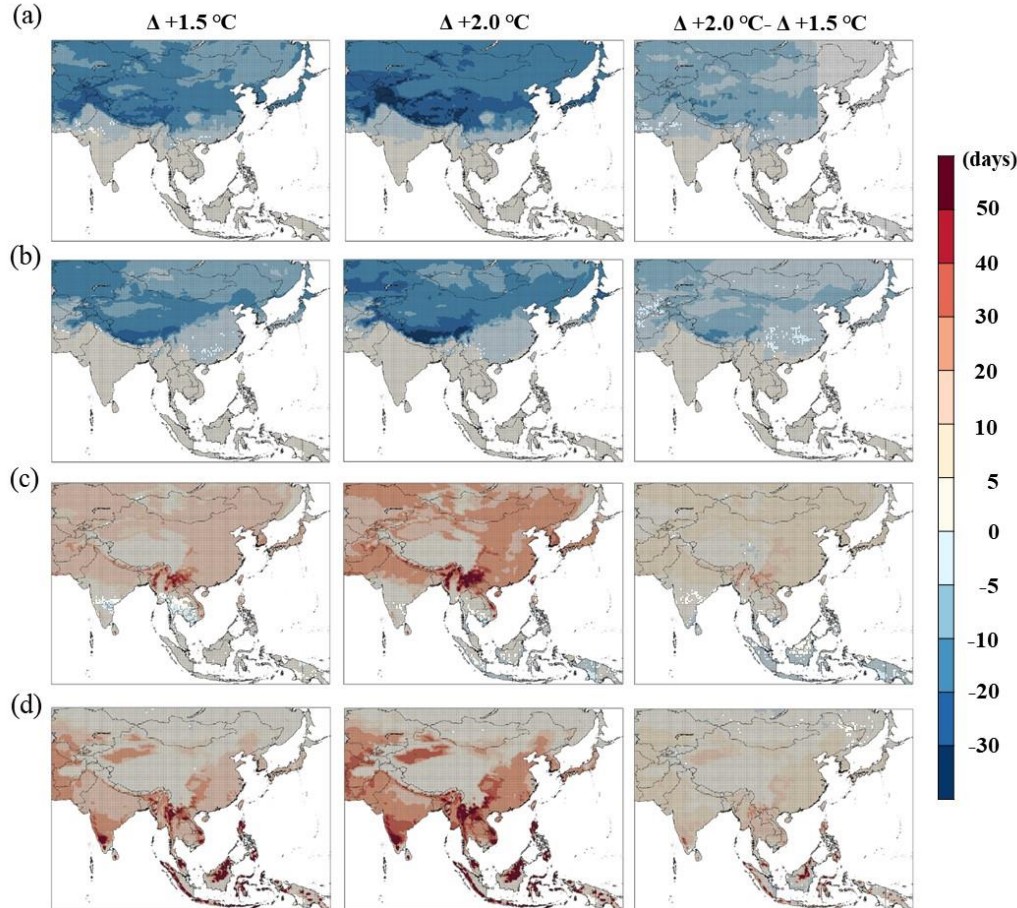

**Figure 4: Spatial distributions of the MME for four extreme climate indices (FD, ID, SU, and TR) over the study domain. The relative changes in the numbers of (a) frost days (FD), (b) ice days (ID), (c) summer days (SU), and (d) tropical nights (TR) for 1.5 °C and 2.0 °C of global warming under RCP4.5 are compared with those of the reference period (REF).**

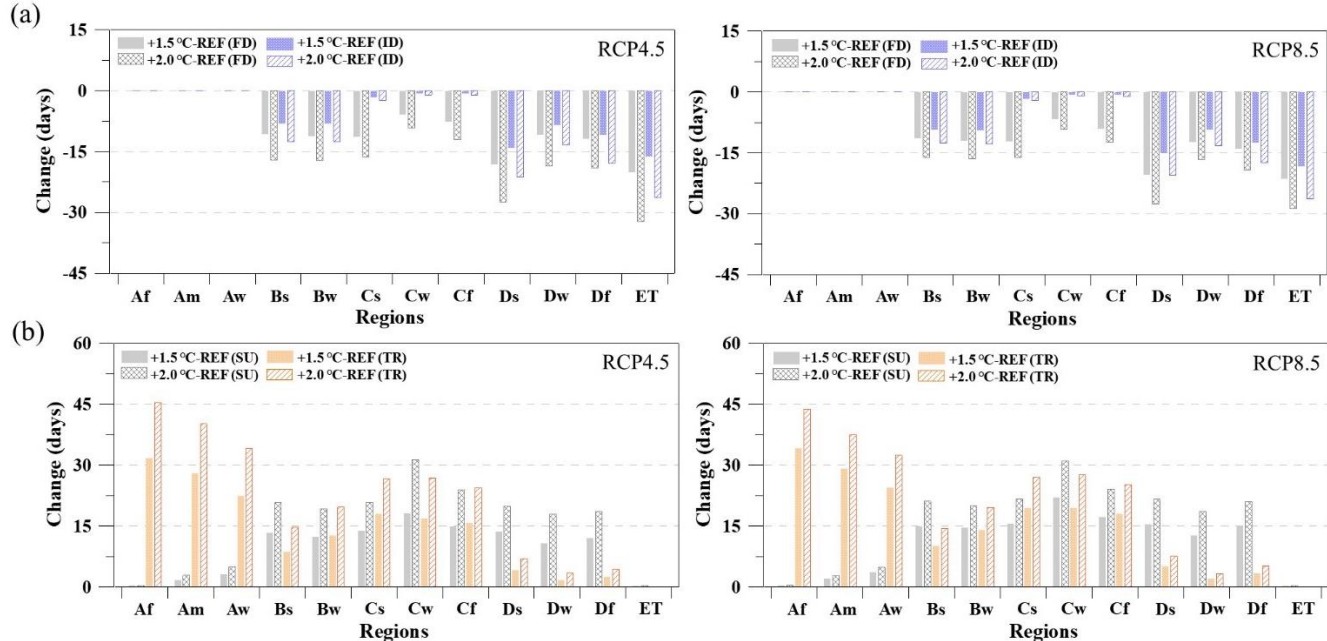

**Figure 5: Changes in the extreme temperature indices (a) related to the coldest days (FD: frost days and ID: ice days) and (b) related to the hottest days (SU: summer days and TR: tropical nights) derived from the MME for the 12 climate zones.**

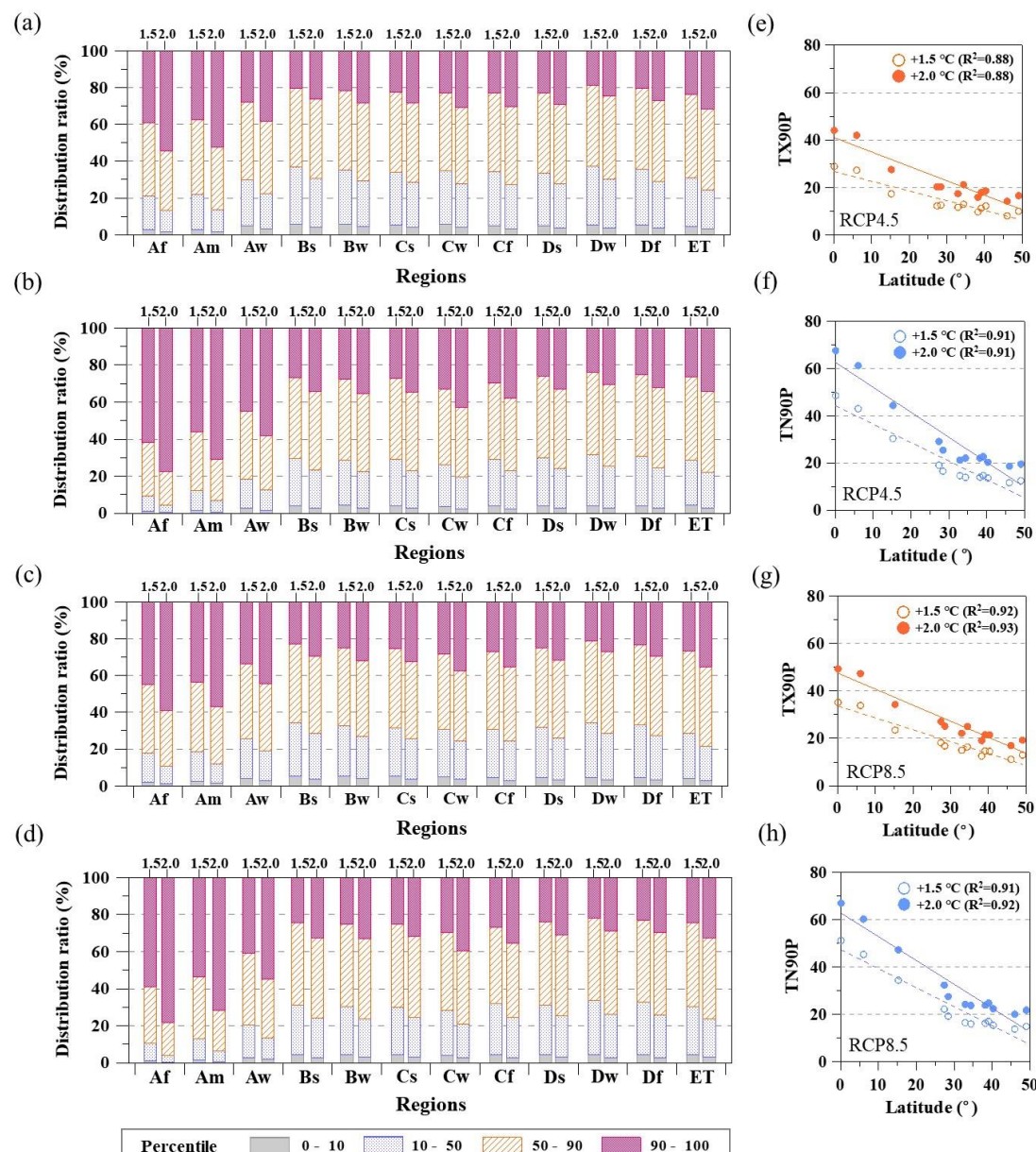

**Figure 6: Relative changes in the (a, c) maximum temperature (TX) and (b, d) minimum temperature (TN) according to four percentile ranges under 1.5 and 2.0 °C of global warming compared with the reference period (REF). (e, g) The relationship between the average latitude of each climate zone and the relative change in the 90th percentile of the maximum temperature (TX90P) under global warming. (f, h) The relationship between the average latitude of each climate zone and the relative change in the 90th percentile of the minimum temperature (TN90P) under global warming. Each circle in (e-h) denotes a representative value for an individual climate zone. The solid (dashed) line in (e-h) represents the regression relationship between two variables with the coefficient of determination ($R^2$) under 1.5 °C (2.0 °C) of global warming.**

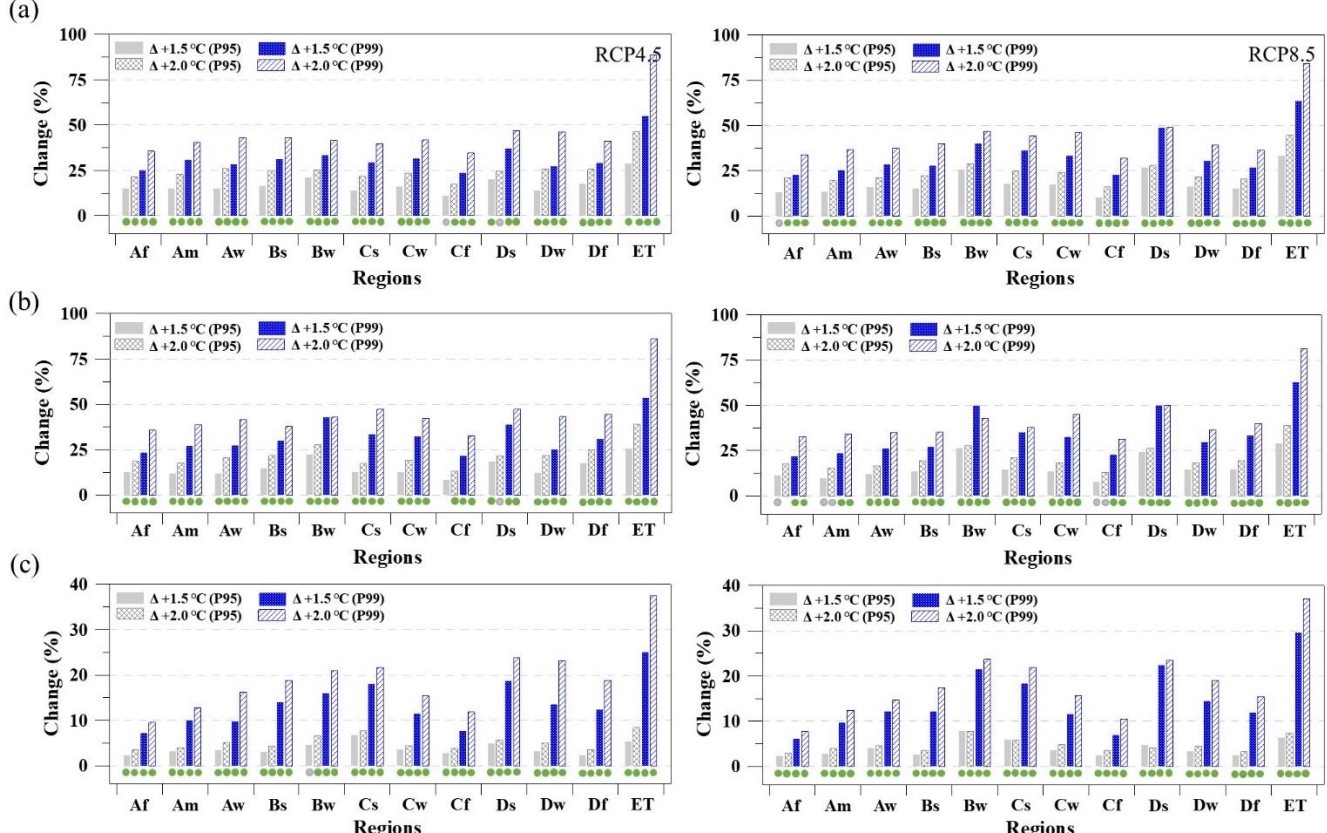

**Figure 7:** Relative changes in the (a) total amount, (b) frequency, and (c) intensity of the extreme precipitation indices (P95: very wet day precipitation, P99: extreme wet day precipitation) for the 12 climate zones derived from the MME of the five GCMs under 1.5 °C and 2.0 °C of global warming compared to the reference period (REF). Green circles (gray circles) denote 100 % (over 80 %) intermodel agreement.

770

(a)

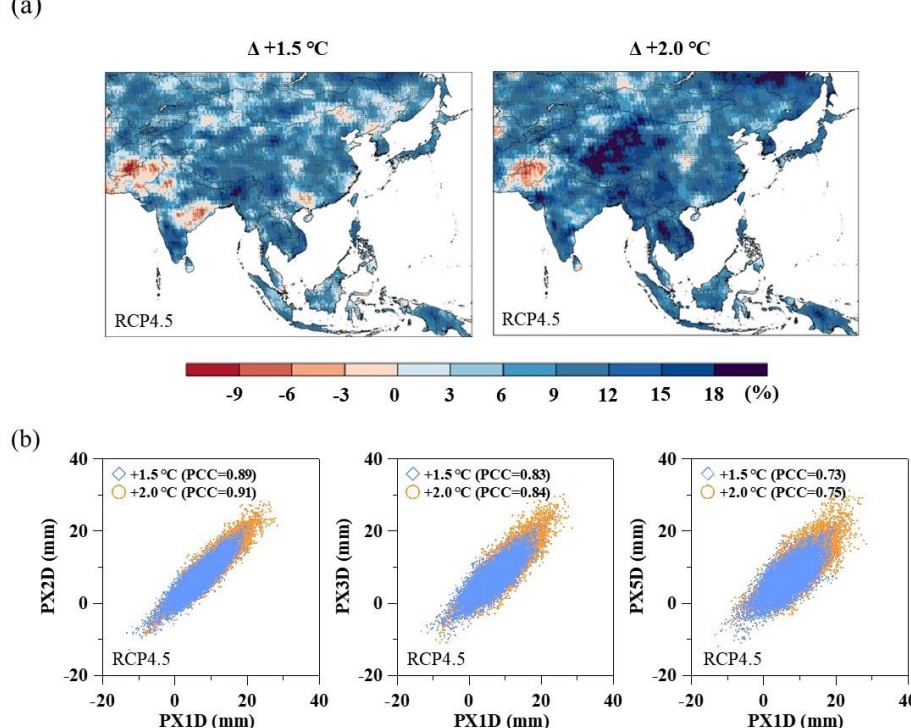

Figure 8: (a) Relative changes in the annual maximum precipitation (PX1D) derived from the MME of the five GCMs under 1.5 °C and 2.0 °C of global warming forced by RCP4.5 compared to the reference period (REF). (b) Scatter plot for a comparison of the relative changes between the maximum precipitation over 2, 3, and 5 consecutive days (PX2D, PX3D, and PX5D, respectively) and the annual maximum precipitation (PX1D) derived from the MME over the Asian monsoon region. Each blue diamond (orange circle) in (b) indicates the relationship between the variable on the x-axis and the variable on the y-axis for an individual grid value within the region under 1.5 °C (2.0 °C) of global warming.

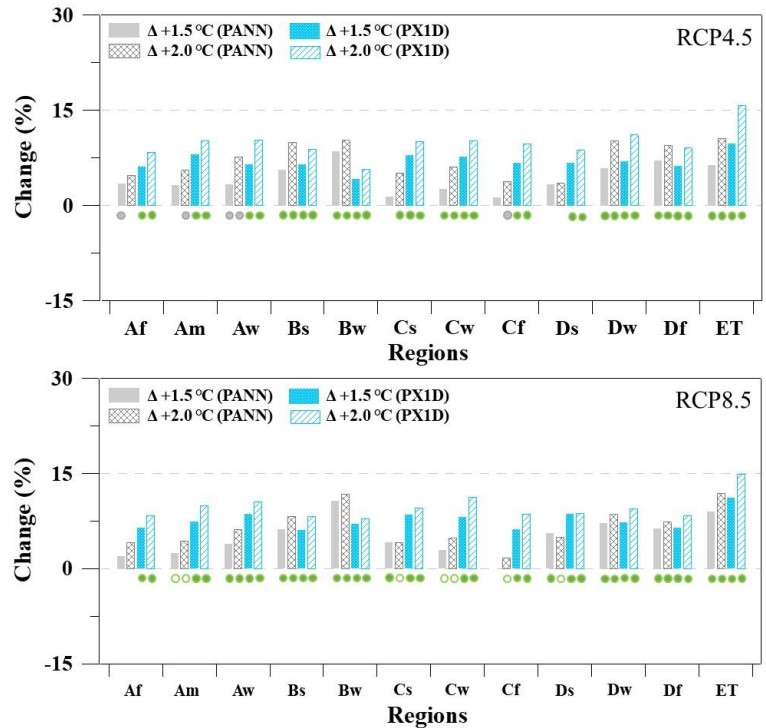

**Figure 9: Relative changes in the annual mean precipitation (PANN) and annual maximum precipitation (PX1D) for the 12 climate zones derived from the MME of the five GCMs under 1.5 °C and 2.0 °C of global warming compared to the reference period (REF). Green circles (gray circles) denote 100 % (over 80 %) intermodel agreement.**

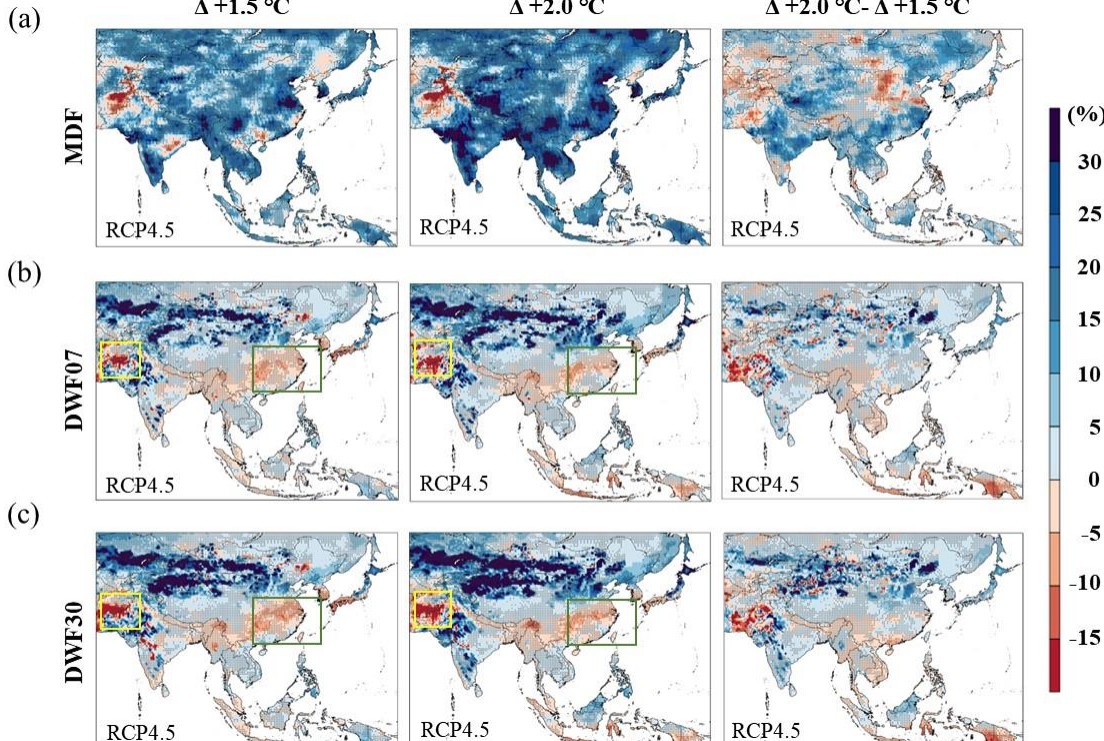

**Figure 10: Spatial distributions of the MME for three extreme climate indices (MDF, DWF07, and DWF30) under RCP4.5 over the study domain. Relative changes in the (a) annual maximum runoff (MDF), (b) consecutive 7-day minimum runoff (DWF07), and (c) consecutive 30-day minimum runoff (DWF30) under 1.5 °C and 2.0 °C of global warming compared to the reference period (REF). The green (i.e., located in the Cf zone) and yellow (i.e., located in the Bw zone) rectangles indicate the locations of regions susceptible to DWF07 and DWF30 under global warming.**

790

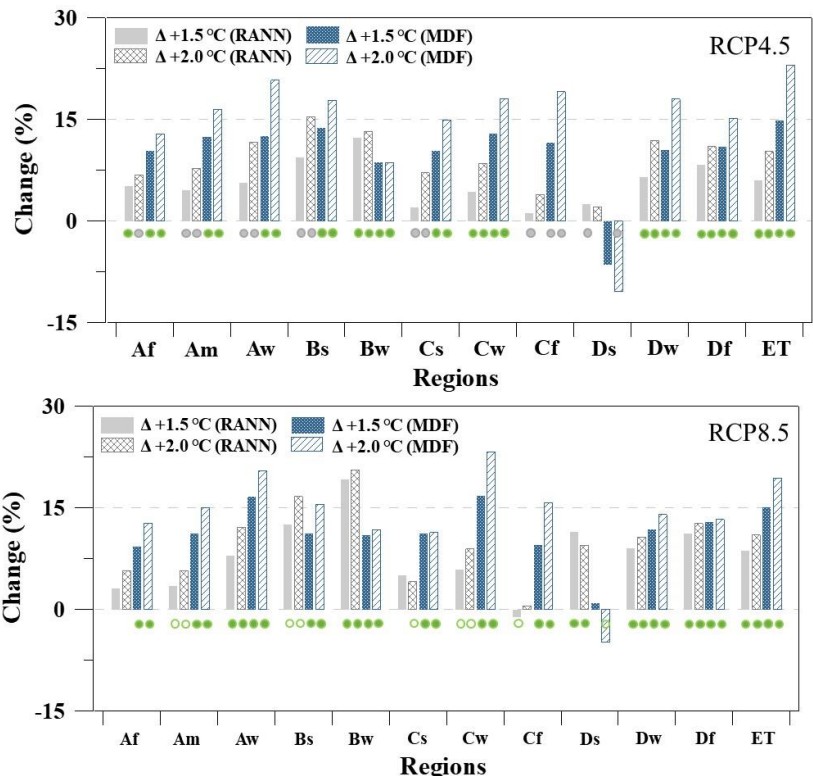

Figure 11: Relative changes in the (a) annual mean runoff (RANN) and (b) annual maximum runoff (MDF) for the 12 climate zones derived from the MME of the five GCMs under 1.5 °C and 2.0 °C of global warming compared to the reference period (REF). Green circles (gray circles) denote 100 % (over 80 %) intermodel agreement.

**Table 1: Climate zones classified according to the Köppen climate classification method using temperature and precipitation (Tmin(max): monthly averaged minimum (maximum) temperature, Pmin: monthly averaged minimum precipitation, PANN: annual averaged precipitation, Psmin(smax): minimum (maximum) precipitation in the summer season, and Pwmin(wmax): minimum (maximum) precipitation in the winter season).**

| Type | Description | Criterion | Ratio of Area (%) |
|------|-------------|-----------|-------------------|
| **A** | **Tropical climates** | $\mathbf{T_{min} \geq +18\ ^\circ C}$ | **17.2** |
| Af | Rainforest | $P_{min} \geq 60$ mm | 5.8 |
| Am | Monsoon | Not(Af) & $P_{min} \geq 100\text{-}P_{ANN}/25$ | 1.7 |
| Aw | Savannah | Not(Af) & $P_{min} < 100\text{-}P_{ANN}/25$ | 9.7 |
| **B** | **Arid climates** | $\mathbf{P_{ANN} < 10P_{th}}$ | **22.1** |
| BS | Steppe climate | $P_{ANN} > 5P_{th}$ | 4.6 |
| BW | Desert climate | $P_{ANN} \leq 5P_{th}$ | 17.5 |
| **C** | **Warm temperate climates** | $\mathbf{-3\ ^\circ C < T_{min} < +18\ ^\circ C}$ | **19.4** |
| Cs | Warm temperate climate with dry summer | $P_{smin} < P_{wmin}$ & $P_{wmax} > 3P_{smin}$ & $P_{smin} < 40$ mm | 2.1 |
| Cw | Warm temperate climate with dry winter | $P_{wmin} < P_{smin}$ & $P_{smax} > 10P_{wmin}$ | 10.5 |
| Cf | Warm temperate climate without dry season | Neither Cs nor Cw | 6.7 |
| **D** | **Cold climates** | $\mathbf{T_{min} \leq -3\ ^\circ C}$ | **36.3** |
| Ds | Cold climate with dry summer | $P_{smin} < P_{wmin}$ & $P_{wmax} > 3P_{smin}$ & $P_{smin} < 40$ mm | 1.8 |
| Dw | Cold climate with dry winter | $P_{wmin} < P_{smin}$ & $P_{smax} > 10P_{wmin}$ | 19.7 |
| Df | Cold climate without dry season | Neither Ds nor Dw | 14.8 |
| **E** | **Polar climates** | $\mathbf{T_{max} < +10\ ^\circ C}$ | **5.1** |
| ET | Tundra climate | $0\ ^\circ C \leq T_{max} < +10\ ^\circ C$ | 5.1 |
| EF | Frost climate | $T_{max} < 0\ ^\circ C$ | - |

**Table 2: List of the five selected GCMs used in this study.**

| No. | GCM | Resolution (Lon.×Lat.) | Institute | Nation |
|-----|-----|------------------------|-----------|--------|
| 1 | bcc-csm1-1-m | $1.125° \times 1.125°$ | BCC | China |
| 2 | CanESM2 | $2.8125° \times 2.8125°$ | CCCma | Canada |
| 3 | CMCC-CMS | $1.875° \times 1.875°$ | CMCC | Italy |
| 4 | CNRM-CM5 | $1.40625° \times 1.40625°$ | CNRM-CERFACS | France |
| 5 | NorESM1-M | $2.5° \times 1.875°$ | NCC | Norway |

**Table 3: Central years (corresponding periods) of the individual GCMs with global warming of 0.48 ℃, 1.5 ℃, and 2.0 ℃ under the RCP4.5 and RCP8.5 scenarios.**

| No. | GCM | Reference period (0.48 ℃) | RCP4.5 scenario | | RCP8.5 scenario | |
|---|---|---|---|---|---|---|
| | | | Future period (1.5 ℃) | Future period (2.0 ℃) | Future period (1.5 ℃) | Future period (2.0 ℃) |
| 1 | bcc-csm1-1-m | 1973 (1959-1988) | 2013 (2006-2035) | 2039 (2025-2054) | 2012 (2006-2035) | 2030 (2016-2045) |
| 2 | CanESM2 | 1983 (1969-1998) | 2016 (2006-2035) | 2031 (2017-2046) | 2012 (2006-2035) | 2026 (2012-2041) |
| 3 | CMCC-CMS | 1996 (1982-2011) | 2034 (2020-2049) | 2052 (2038-2067) | 2030 (2016-2045) | 2040 (2026-2055) |
| 4 | CNRM-CM5 | 1988 (1974-2003) | 2035 (2021-2050) | 2056 (2042-2071) | 2029 (2015-2044) | 2043 (2029-2058) |
| 5 | NorESM1-M | 1991 (1977-2006) | 2041 (2027-2056) | 2075 (2061-2090) | 2033 (2019-2048) | 2048 (2034-2063) |

815

**Table 4: Definitions of the hydroclimatic extreme indices using minimum temperature (denoted by TN), maximum temperature (denoted by TX), precipitation (denoted by PR), and runoff data, where 'i' and 'j' represent the month and year, respectively.**

| Index name (label) | Index definition | Unit | Source of indices |
|---|---|---|---|
| Tropical nights (TR) | The number of days when TNij > 20 ℃ | Days | Minimum temperature |
| Frost days (FD) | The number of days when TNij < 0 ℃ | Days | |
| Warm nights (TN90P) | The number of days when TNij > TNref90; here, TNref90 is the calendar day 90th percentile centered on a 5-day window for the reference period of individual GCMs | Days | |
| Cold nights (TN10P) | The number of days when TNij < TNref10; here, TNref10 is the calendar day 10th percentile centered on a 5-day window for the reference period of individual GCMs | Days | |
| Summer days (SU) | The number of days when TXij > 25 ℃ | Days | Maximum temperature |
| Ice days (ID) | The number of days when TXij < 0 ℃ | Days | |
| Warm days (TX90P) | The number of days when TXij > TXref90; here, TXref90 is the calendar day 90th percentile centered on a 5-day window for the reference period of individual GCMs | Days | |
| Cold days (TX10P) | The number of days when TXij < TXref10; here, TXref10 is the calendar day 10th percentile centered on a 5-day window for the reference period of individual GCMs | Days | |
| Very wet day precipitation (P95) | The total precipitation when PRij exceeds the 95th percentile of the wet day precipitation in the reference period of individual GCMs | Mm | Precipitation |
| Extreme wet day precipitation (P99) | The total precipitation when PRij exceeds the 99th percentile of the wet day precipitation in the reference period of individual GCMs | Mm | |
| Annual maximum precipitation (PX1D) | The maximum 1-day precipitation | Mm | |
| Maximum 2-day precipitation (PX2D) | The maximum consecutive 2-day precipitation | Mm | |
| Maximum 3-day precipitation (PX3D) | The maximum consecutive 3-day precipitation | Mm | |
| Maximum 5-day precipitation (PX5D) | The maximum consecutive 5-day precipitation | Mm | |
| Minimum 7-day runoff (DWF07) | The minimum consecutive 7-day runoff | Mm | Runoff |
| Minimum 30-day runoff (DWF30) | The minimum consecutive 30-day runoff | Mm | |
| Annual maximum runoff (MDF) | The maximum daily runoff | Mm | |

820

**Table 5: Plots of the percentage changes (%) in the climate extreme indices in response to additional warming of 0.5 °C in the climate zones over Asia under (a) RCP4.5 and (b) RCP8.5, where '*' and '**' represent significance at the 80 and 100 % levels, respectively.**

(a)

| | FD | ID | SU | TR | TX10P | TN10P | TX90P | TN90P | PANN | PX1D | PX2D | PX3D | PX5D | P95 | P99 | RANN | MDF | DWF07 | DWF30 |
|---|---|---|---|---|---|---|---|---|---|---|---|---|---|---|---|---|---|---|---|
| Af | - | - | - | ** | ** | ** | ** | ** | * | ** | ** | ** | ** | * | | * | | | |
| Am | - | - | ** | ** | ** | ** | ** | ** | * | * | ** | ** | ** | ** | ** | * | * | | |
| Aw | - | - | ** | ** | ** | ** | ** | ** | ** | * | * | * | * | * | * | ** | * | | |
| BS | ** | ** | ** | ** | ** | ** | ** | ** | ** | * | * | ** | ** | ** | * | ** | * | | |
| BW | ** | ** | ** | ** | ** | ** | ** | ** | | * | * | * | * | | * | | * | | |
| Cs | ** | ** | ** | ** | ** | ** | ** | ** | | * | * | | | | | | * | | |
| Cw | ** | ** | ** | ** | ** | ** | ** | ** | ** | * | ** | ** | * | ** | * | ** | ** | | |
| Cf | ** | ** | ** | ** | ** | ** | ** | ** | * | * | ** | ** | ** | * | ** | * | ** | | |
| Ds | ** | ** | ** | ** | ** | ** | ** | ** | | * | * | * | * | | | | * | | |
| Dw | ** | ** | ** | ** | ** | ** | ** | ** | ** | ** | ** | ** | ** | ** | ** | ** | ** | | |
| Df | ** | ** | ** | ** | ** | ** | ** | ** | ** | ** | ** | ** | ** | ** | ** | ** | ** | | |
| ET | ** | ** | ** | - | ** | ** | ** | ** | ** | ** | ** | ** | ** | ** | ** | ** | ** | | * |

(b)

| | FD | ID | SU | TR | TX10P | TN10P | TX90P | TN90P | PANN | PX1D | PX2D | PX3D | PX5D | P95 | P99 | RANN | MDF | DWF07 | DWF30 |
|---|---|---|---|---|---|---|---|---|---|---|---|---|---|---|---|---|---|---|---|
| Af | - | - | - | ** | ** | ** | ** | ** | * | ** | ** | ** | ** | * | ** | * | ** | | |
| Am | - | - | ** | ** | ** | ** | ** | ** | * | ** | ** | ** | ** | ** | ** | ** | ** | | |
| Aw | - | - | ** | ** | ** | ** | ** | ** | * | ** | ** | ** | * | ** | ** | ** | ** | * | * |
| BS | ** | ** | ** | ** | ** | ** | ** | ** | * | * | * | * | * | ** | ** | * | * | | * |
| BW | ** | ** | ** | ** | ** | ** | ** | ** | | | | | | | * | | | | * |
| Cs | ** | ** | ** | ** | ** | ** | ** | ** | | | | | | * | * | | | * | * |
| Cw | ** | ** | ** | ** | ** | ** | ** | ** | ** | ** | ** | ** | ** | ** | ** | * | ** | | * |
| Cf | ** | ** | ** | ** | ** | ** | ** | ** | | * | ** | ** | | | * | | * | | |
| Ds | ** | ** | ** | ** | ** | ** | ** | ** | | | | | | | | ** | | | * |
| Dw | ** | ** | ** | ** | ** | ** | ** | ** | ** | ** | ** | * | ** | ** | ** | ** | * | | |
| Df | ** | ** | ** | ** | ** | ** | ** | ** | * | ** | ** | * | ** | ** | ** | * | | ** | ** |
| ET | ** | ** | ** | - | ** | ** | ** | ** | ** | ** | ** | ** | ** | ** | ** | ** | ** | * | * |

-20  -15  -10  -8  -6  -4  -2  0  2  4  6  8  10  15  20  (%)