# Peer review of "Intensification characteristics of hydroclimatic extremes in the Asian monsoon region under 1.5 and 2.0 °C of global warming"

_Hydrology and Earth System Sciences, 2019_

## Referee Comment (RC1) · Harsh Shah (Referee) · 22 Feb 2020

This manuscript highlights the projection of hydroclimatic extremes in the Asia monsoon region under global warming. The authors have comprehensively examined seventeen indices using the best five GCMs over different climatic zones in Asia. However, I feel substantial work to be done to improve the paper. Hence, I request an editor to give more time for the revision of the article. I would love to review the revised draft, and the authors can find my major and minor comments/suggestions below:

[Figure]

Major:

1. No robust finding is coming out from the abstract. It contains a few lines of introduction, method, and overall results. The abstract has to be crisp, short, and quantitative with results.

2. What are the major scientific questions of study?

3. How the selection of GCMs varies for the different climatic zone? And what are RMSE and SSC range to assign a score from -1, 0, 1? Why is the score given based on MME not against observed? MME comparison may induce bias, and I recommend to re-calculate GCMs performance (calculate Pbias) against observed data for each climatic zone. To see selections of GCMs remain the same for all climatic zone or not? Also, I would like to see selected five GCMs performance against observed data from the 2006-2019 period.

4. Why is bias correction applied after the selection of reference period for individual GCMs? How bias corrections affect the 1.5 and 2 °C central years for reference?

5. How APHRODITE and University of Washington data could perform against CERA-20C reanalysis?

6. Make a similar plot (Figure 2) using observed and MME data for temperature and simulated runoff. You can keep those in the supplemental material.

7. The figure's quality is poor.

Minor:

Line 25: "increased the frequency and intensity of natural disasters" Give citations.

Line 71-77: I recommend to keep these in data and method sections—no need to brief in the introduction.

Line 81: "As far as we know, relatively few studies" Give citations.

[Figure]

Line 120-123: How to relate 1.5-2.0 °C global warming with the RCP4.5 scenario? The small description of this will help the reader to understand the process.

Line 136: What is a central year for reference for MME at 0.48, 1.5, and 2 °C?

Line 137-138: How come the central year of PI is 1895? When your PI period is 1861-1890? Please check.

Line: 155: provide the full name of VIC.

Keep observational datasets section after section 2.1

Line 197: what are the reference period and two future periods? Is it for individual GCM or MME? Please clear to avoid confusion.

Table 4: Also, provide a source of indices (e.g., minimum temperature, maximum temperature, precipitation, and runoff) in another column.

Section 3.1: Is climatic zone classification is based on observed data? And which year? Also, mention which observation reanalysis or APHRODITE and University Washington data?

Line 219: "the bias-corrected GCMs are validated" Justify this result for temperature as well.

---

## Referee Comment (RC2) · Anonymous Referee #2 · 5 Apr 2020

Overview

The manuscript presents an analysis of likely changes in future temperature, precipitation, and runoff extremes over different climatic regions of Asia. Climate projections are obtained from a suite of climate models and a hydrologic model is used to translate climate to runoff. While the manuscript addresses a topic of relevance to the journal, it has several major shortcomings that prohibit readers from interpreting the results and gauging their reliability. These are listed below:

[Figure]

1. Novelty: There are several studies that discuss the consequences of 1.5°C and 2.0°C global warming on hydrology of river basins across the globe. A number of such analyses are cited in the paper but some significant research is not discussed. For example Betts et al. (2018) discuss the difference between the two global warming levels in terms of hydrologic extremes as well as food security. Similarly, Doll et al. (2018) employ two hydrologic models and an ensemble of bias corrected climate models to understand how freshwater related hazards are likely to change across the globe under the two warming levels. A number of similar studies can be found. The authors need to justify how their analysis adds value to this literature. A statistical analysis of expected changes is useful but these numbers need to be eventually translated into variables that have direct impact on society (such as food availability, flood hazard, etc.). Perhaps the authors can provide some policy relevant insights to the readers, for example, by suggesting how the adaptive measures will vary across different climate regions.

2. Methodology: There are a number of issues here:

a. Climate data: the analysis involves a number of steps and the text is a little hard to follow in this regard. For example, to select the five GCMs, a comparison with a multi-model mean is carried out (Page 3, line 117). But why are GCMs selected on the basis of their performance w.r.t the ensemble mean? Why not directly compare the individual GCM performance with the observed data and select those that best represent the observed climate in the Asia region? It is later revealed that a bias-correction is also carried out on the climate data. Was this bias correction carried out before or after the shortlisting of the five GCMs? Overall, the sequence of methods is unclear and many methodological choices are not defended well in the text. Maybe a flowchart to guide the readers through the main steps will help.

b. Hydrologic model and runoff projections: the variable infiltration capacity model is a commonly used model to obtain continental to global scale runoff projections. However, many studies limit their analysis to understanding patterns in mean annual runoff. In this study, however, the focus is on runoff extremes, which are inherently harder to

capture than a long-term mean value. However, the calibration and validation of the model are not included in the main manuscript. A regionalization approach is used to transfer parameters from gauged to ungauged sites, but how successful was this? It is important to show that the model with the regionalization scheme can capture the observed hydrologic extremes in the past. Only then, we can have some confidence regarding the reliability of the projections in the future.

c. Choice of extreme indices: The choice of indices for precipitation and runoff seem counter-intuitive. The precipitation indices focus on only high precipitation events while the runoff indices focus on both high and low events. Why not include precipitation extremes that involve minimum or very low precipitation?

d. Selection of time periods: A time sampling method is used to identify the time period of analysis for various GCMs. The authors arrive at single time period for each GCM-warming level. This suggests that a spatially aggregated value of climate indices was used to identify the time periods. However, the analysis focuses on different climate regions and it is possible that each climate region reaches a global warming level in different time periods. Why was this spatial heterogeneity ignored? The same applies on the bias correction methodology, which could have been applied on each homogeneous climate region one by one. On a similar note, the climate zone classification results are presented in Line 206 onwards. Is this classification carried out using observed data or GCM data? How sensitive is the classification to the choice of climate data?

e. Choice of scenarios: it is not clear why RCP4.5 was chosen for the analysis when RCP6.0 and RCP8.5 are equally relevant.

3. Presentation: Overall, the manuscript can gain from improvement in language. In addition, the figure clarity can be improved. The figure captions are not very descriptive and it is hard to follow what is on the figures without carefully reading the main text. Please explain all symbols and abbreviations used in the figures in the caption itself.
References Betts, Richard A., Lorenzo Alfieri, Catherine Bradshaw, John Caesar, Luc Feyen, Pierre Friedlingstein, Laila Gohar et al. "Changes in climate extremes, fresh water availability and vulnerability to food insecurity projected at 1.5 C and 2 C global warming with a higher-resolution global climate model." Philosophical Transactions of the Royal Society A: Mathematical, Physical and Engineering Sciences 376, no. 2119 (2018): 20160452. Döll, Petra, Tim Trautmann, Dieter Gerten, Hannes Müller Schmied, Sebastian Ostberg, Fahad Saaed, and Carl-Friedrich Schleussner. "Risks for the global freshwater system at 1.5 C and 2 C global warming." Environmental Research Letters 13, no. 4 (2018): 044038.

---

## Author Comment (AC1) · 3 May 2020

Please see of supplementary file.

Please also note the supplement to this comment:
https://www.hydrol-earth-syst-sci-discuss.net/hess-2019-643/hess-2019-643-AC1-supplement.zip
* * *
[Figure]

643, 2020.

---

## Author Comment (AC2) · 3 May 2020

Please see of supplementary file.

Please also note the supplement to this comment:
https://www.hydrol-earth-syst-sci-discuss.net/hess-2019-643/hess-2019-643-AC2-supplement.zip

643, 2020.

---

## Author Response (AR1)

Dear Editor and Reviewers:

First, we would like to thank the editor and reviewers for their helpful comments and suggestions, which improved the quality of our manuscript. We agree with most of the concerns raised by the reviewers and have therefore modified the manuscript according to the reviewers' comments and suggestions. Newly added and modified text is highlighted in yellow in the revised manuscript, and our point-by-point responses to the reviewers' comments are provided below. We hope that the revised manuscript is now suitable for publication in Hydrology and Earth System Sciences.

**Reply to Reviewer (#1)'s Comments:**

*This manuscript highlights the projection of hydroclimatic extremes in the Asia monsoon region under global warming. The authors have comprehensively examined seventeen indices using the best five GCMs over different climatic zones in Asia. However, I feel substantial work to be done to improve the paper. Hence, I request an editor to give more time for the revision of the article. I would love to review the revised draft, and the authors can find my major and minor comments/suggestions below:*

► We appreciate the reviewer's feedback and helpful comments. Kindly find our detailed response to each comment below.

*Major:*
*1. No robust finding is coming out from the abstract. It contains a few lines of introduction, method, and overall results. The abstract has to be crisp, short, and quantitative with results.*

► We have revised the abstract considering the reviewer's comment.

: The influences of global warming have contributed to changes in hydroclimatic extremes, which are more complicated at the regional scale. To reduce the potential risk of extremes under future climate, assessing the change of extreme climate events is important, especially in Asia due to various different climates features and seasonal variability. Here, the changes in hydroclimatic extremes are assessed over the Asian monsoon region under global mean temperature warming targets of 1.5 and 2.0 ℃ above preindustrial levels based on representative concentration pathways (RCP) 4.5 and 8.5. Analyses of the subregions classified using regional climate characteristics are performed based on the multimodel ensemble mean (MME) of five bias-corrected global climate models (GCMs). For runoff extremes, the hydrologic responses to 1.5 and 2.0 ℃ global warming targets are simulated based on the variable infiltration capacity (VIC) model. The temperature extremes show significant change patterns over all climate zones. As the globe warms, increasing warm extremes (fewer than 45 days) and decreasing cold extremes (fewer than 32 days) occur more frequently over Asia with strong robustness. Moreover, changes in precipitation and runoff averages (and low runoff extremes) show change patterns with large spatial variations featuring weak robustness based on intermodel agreement. Additionally, global warming is expected to significantly intensify maximum precipitation extremes (usually exceeding a 10 % increase in intensity under 2.0 ℃ of warming) in all climate zones. Regardless of regional climate characteristics and RCPs, this behavior is expected to be enhanced under the 2.0 ℃ (compared with the 1.5 ℃) warming scenario and increase the likelihood of flood risk (up to 10 %). Additionally, the spatial extent and magnitude of the runoff change patterns are modulated by those of the precipitation change patterns. More importantly, an extra 0.5 ℃ of global warming under 2 RCPs will amplify the change

patterns of hydroclimatic extremes with strong robustness, especially in cold (and polar) climate zones. The results of this study clearly demonstrate the changes patterns in regional hydroclimatic extremes (e.g., temperature and high precipitation) under warmer conditions over Asia and confirm that hydroclimatic sensitivities differ based on regional climate characteristics.

45

*2. What are the major scientific questions of study?*

► Thank you for this comment. The major scientific question of this study is to examine how hydroclimatic extremes respond differently in regions with unique climate features under global warming of 1.5 and 2.0 ℃. We provide this text in "Section 1. Introduction".

50 : Because the impacts of global temperature increases impact each region separately due to the changes in regional climate features, the hydroclimatic changes in response to global warming reflect unique regional responses. Hence, global warming leads to changes in regional hydroclimatic extremes according to regional sensitivities, which remain uncertain. Therefore, the main purposes of this study are to examine the potential impacts of regional climate on hydroclimatic extremes under different global warming conditions and to investigate the regional-scale sensitivity of individual hydroclimatic variables

55 to increases in the global mean temperature with diverse climate features.

*3. How the selection of GCMs varies for the different climatic zone? And what are RMSE and SSC range to assign a score from -1, 0, 1? Why is the score given based on MME not against observed? MME comparison may induce bias, and I recommend to re-calculate GCMs performance (calculate Pbias) against observed data for each climatic zone. To see*

60 *selections of GCMs remain the same for all climatic zone or not? Also, I would like to see selected five GCMs performance against observed data from the 2006-2019 period.*

► The selection of GCMs is the same within the study area (i.e., the Asian monsoon region) regardless of the climate zone because we select the best-performing GCMs to simulate the spatial patterns of Asian climate features compared with observations. We have clarified this point in the revised

65 manuscript.

: Reliable climate change scenarios, which are derived from the selected GCMs, are important sources for estimating the impacts of global warming on hydroclimatic (e.g., temperature, precipitation, and runoff) extremes. Here, the method for selecting GCMs suggested by Kim et al. (2020) is employed while focusing on their performance in simulating the spatial patterns of observed climate features in Asia because the regional climate is affected by physical climate system processes

70 that occur over large spatial scales (e.g., from the planetary scale to the synoptic scale and mesoscale). For the future projections, the selected GCMs are applied to the entire domain regardless of the climate zone.

► For the second comment, the RMSE and SCC criteria used to assign scores for the 12 individual climate variables range from 0.0 (specific humidity) to 409.9 (geopotential height) for RMSE and from 0.37 (geopotential height) to 0.94 (near-surface air mean temperature) for SCC. This shows the large

75 variability among the climate variables because the values of the criteria depend strongly on the modeling uncertainty existing in the simulation of individual climate variables.

► Here, we apply the MME-based scoring rule to exclude the low-performing GCMs and select the best-performing GCMs based on the relative concept. The individual GCM performance in this rule is judged by comparison with the MME performance, where the MME is considered the score under the

80 assumption that the MME is similar to the observed data compared with each GCM (Xu et al., 2020;

Tegegne et al., 2020). Moreover, the scoring rule based on observed data can be used to determine the rank of the GCMs; however, this rule does not provide the information needed to screen the GCMs. We clarified why we apply the MME-based scoring rule in the revised manuscript as follows:

: Next, we apply the MME-based scoring rule for the section of GCMs (Nyunt et al., 2012) to exclude low-performing GCMs
85   and select only the best-performing GCMs by using a relative concept because the scoring rule based on the observed data does not provide the information needed to screen the GCMs. Therefore, the individual GCM statistics (i.e., the SCC and RMSE) are judged by comparison with the MME statistic. The value of MME statistics are considered as criteria to give a score to each GCM under the assumption that the MME is similar to the observed data compared with the output from only one GCM (Xu et al., 2020; Tegegne et al., 2020).

90   ► Also, the root mean square error (RMSE) and spatial correlation coefficient (SCC) between each GCM simulation field and observed data field are used in this study to examine the GCM performance in simulating the observed spatial climate features. Of course, the GCM performance can be evaluated from the Pbias, but we applied the RMSE statistic instead of Pbias to measure the error between the simulation and observation. Since the RMSE and SCC are commonly used to validate GCM
95   simulations (IPCC, 2007; McSweenet et al., 2015), we hope this approach is acceptable. We clarified the explanation of the GCM selection approach in the revised manuscript.

: The spatial correlation coefficient (SCC) and root mean square error (RMSE) between the historical simulation field derived from each GCM and the observed field are calculated for each of the twelve relevant variables over the Asian monsoon region, as these statistics are commonly used to examine the performance of GCMs in the simulation of observed spatial
100   climate features (IPCC, 2013; McSweeney et al., 2015).

► And, the selected GCMs are employed for all climatic zones as suggested in the response of the first comment.

► For the final comment, we understand the concern regarding the validation of the GCMs raised by the reviewer. However, the future simulation period of the AR5 GCM started in 2006 based on
105   representative concentration pathway (RCP) scenarios. Although the RCPs are consistent with a wide range of possible changes in future anthropogenic GHG emissions, the RCPs are obviously different from the real world. Since it is difficult to directly compare the selected GCM simulations with observations for the 2006-2019 period, we validated the selected GCMs with the observed data (e.g., mean and maximum values) from a historical period (1976-2005), as shown in Figure 3 for
110   precipitation, Figure S5 for temperature, and Figure S6 for runoff.

: The validation results of the MME compared with the OBS for the minimum (and maximum) temperature are illustrated in Figure S5. The MME outputs of minimum and maximum temperature are very similar to the OBS temperature patterns. In addition, the simulated runoff based on the MME and OBS are compared due to the lack of measured runoff data (Figure S6). The MME results show reasonable historical simulations with implications for the reliability of the climatological and
115   hydrological responses to the climate forcing derived from the MME.

[Figure]

Figure 3: Spatial distributions of the (a) annual mean precipitation (PANN) and (b) annual maximum precipitation (PX1D) for the historical period (1976-2005) in the Asian monsoon region derived from observations and the MME of bias-corrected outputs from the five GCMs.

120

[Figure]

**Figure S5: Spatial distributions of the (a) annual minimum temperature (unit: °C) and (b) annual maximum temperature (unit: °C) for the historical period (1976-2005) in the Asian monsoon region. OBS and MME denote the values obtained from the observational temperature dataset and the MME of bias-corrected outputs from the five GCMs, respectively.**

125

[Figure]

**Figure S6: Spatial distributions of the (a) annual mean runoff (unit: mm) and (b) daily maximum runoff (unit: mm/day) for the historical period (1976-2005) in the Asian monsoon region. OBS denotes the simulated runoff from the VIC model fed by observational precipitation data (i.e., APHRODITE). MME denotes the simulated runoff from the VIC model fed by the MME of the bias-corrected outputs from the five GCMs.**

130

*4. Why is bias correction applied after the selection of reference period for individual GCMs? How bias corrections affect the 1.5 and 2 °C central years for reference?*

► Thank you for this comment. We applied the bias correction method after the determination of both the reference period (i.e., corresponding to 0.48 °C) and two future periods (i.e., corresponding to 1.5 and 2 °C) for each GCM. The purpose of applying a statistical bias correction is to reproduce regional (or local)-scale climate information based on large-scale climate information (i.e., GCM outputs) and observed regional features from climate impact studies, while the reference period and future periods for individual GCMs are defined by identifying global mean temperature responses to global warming targets. Since each period is taken based on the increment in the globally averaged temperature above the PI level, bias corrections are not necessary for this process and do not affect it. We added a flowchart (Figure 2) for the climate change scenario to guide the readers, and we have clarified this explanation in the manuscript.

: **2.3 Methodology**

Figure 2 presents a flowchart of the entire procedure used in the study. To simulate the climate during both historical and future periods, climate projections forced by historical and representative concentration pathways (RCPs) 4.5 and 8.5 are selected. The five of the raw GCMs of the Coupled Model Intercomparison Project Phase 5 (CMIP5; Taylor et al., 2012) are selected by applying a unique evaluation procedure (Kim et al., 2020). Then, a reference 30-year period and two future 30-year periods of individual GCM projections are defined under warming targets of 0.48, 1.5 and 2.0 °C above PI levels (1861-1890) based on a time sampling method. Then, these daily forcing data (e.g., precipitation, maximum temperature, and minimum temperature) are extracted from the five selected GCM projections and then statistically bias-corrected using the quantile mapping method. The bias-corrected GCMs are used as meteorological forcings to run the variable infiltration capacity (VIC) hydrological model. The future changes in the hydroclimatic mean and extremes corresponding to the conditions at warming targets of 1.5 and 2.0 °C are spatially analyzed according to the identified subregions based on climate zones. We focus on the hydroclimatic extreme responses to temperature, precipitation, and runoff variations under global warming targets (i.e., 1.5 and 2.0 °C) using extreme indices. A more detailed description of each procedure is provided in section 2.4, section 2.5, and section 2.6.

[Figure]

**Figure 2: Flowchart of the entire procedure used in this study.**

*5. How APHRODITE and University of Washington data could perform against CERA-20C reanalysis?*

► We collected the observational meteorological dataset considering the data availability for long-term records and the time scale. For the meteorological inputs for the VIC model, we preferentially collected daily observational data (i.e., daily precipitation from APHRODITE and daily minimum and maximum temperatures from the University of Washington). The remaining climate variables were obtained from the CERA-20C reanalysis on a monthly basis due to the limited availability of data. We have clarified this point in the revised manuscript.

: Observational meteorological datasets are required as input variables to the hydrological model on a daily time scale and for validating the performance of the GCM simulations on a monthly time scale. We select the meteorological datasets considering the availability of long-term records and their time scales. To run the hydrological simulations (1950-2005), we collect precipitation data from the Asian Precipitation Highly Resolved Observational Data Integration Toward Evaluation of Water Resources (APHRODITE) product (Yatagai et al., 2012), and the maximum and minimum temperature data and wind speed data are obtained from gridded forcing datasets provided by the University of Washington (Adam and Lettenmaier, 2003; Adam et al., 2006). To evaluate the performance of the GCM simulations, the reanalysis data for the remaining climate variables are obtained from the Coupled European Centre for Medium-Range Weather Forecasts (ECMWF) Reanalysis system-20C (CERA-20C) (Laloyaux et al., 2018) on a monthly basis due to the limited availability of data. These observational datasets including the reanalysis data (hereafter "OBS"), are gridded at a 0.5° spatial resolution and interpolated to the same grid system as the GCMs.

*6. Make a similar plot (Figure 2) using observed and MME data for temperature and simulated runoff. You can keep those in the supplemental material.*

► Thank you for this comment. We have added Figures S5 for temperature and S6 for runoff (as shown above) to the Supplementary Material and revised the manuscript (see response to the final comment No. 3)

*7. The figure's quality is poor.*

► We upgraded the figures to high quality (600 dpi).

*Minor:*

*- Line 25: "increased the frequency and intensity of natural disasters" Give citations.*

► We have added citations.

: The climate system in this region has changed as a result of global warming, and consequently, the frequency and intensity of natural disasters related to climate (e.g., heatwaves, heavy precipitation, and floods) have increased (Thomas et al., 2013; IPCC, 2013; Thomas et al., 2014; Thomas et al., 2015).

*- Line 71-77: I recommend to keep these in data and method sections˘A˘Tno need to brief in the introduction.*

► We have attempted to make the text more concise in the Introduction

: In this study, we assess the changes in climate (and hydroclimatic) extremes corresponding to the warming targets of 1.5 and 2.0 ℃ with a focus on the broad continental-scale climate zones of the Asian monsoon region (Figure 1), as delineated by Bae et al. (2013). Since climate extreme events are an inherent climate component, we classify the subregions in the

Asian monsoon region considering regional climate characteristics to understand the change behaviors of climate (and hydroclimatic) extremes under global warming. To consider the reliability of future projections, we present the results based on the multimodel ensemble mean (MME) derived from five selected GCM projections, including intermodal agreement. This study provides scientific information for policy makers to identify regional patterns of the changes in extremes and thereby recognize the impacts of anthropogenically induced warming.

And, we have suggested detailed information in the Materials and methodology section (section 2.2~2.6).

*- Line 81: "As far as we know, relatively few studies" Give citations.*

► We have added citations.

: To the best of our knowledge, relatively few studies have examined the impacts of global warming on extreme hydroclimatic variable-related responses considering the regional climate in Asia (Liu et al., 2019; Kim et al., 2020; Zhao et al., 2020).

*- Line 120-123: How to relate 1.5-2.0℃ global warming with the RCP4.5 scenario? The small description of this will help the reader to understand the process.*

► This comment is well taken. As the reviewer and editor suggested, we have implemented an additional RCP (i.e., RCP8.5); we have analyzed the results of changes in hydroclimatic extremes under 1.5 and 2 °C of global warming based on two different RCPs in the revise paper (i.e., RCP4.5 and RCP8.5). In this regard, we have added an explanation as follows:

: Our focus is to understand the changes in extreme hydroclimatic conditions under global warming environments of 1.5 and 2.0 °C. The timing to reach specific warming levels for individual GCMs depends on the representative concentration pathway because future projections are forced by these scenarios. The temperature response to different RCPs varies, and therefore, the increasing trend and slope of the global mean temperature differ. Here, the analysis is based on RCP4.5 and RCP8.5, which are commonly considered for realistic future projections. RCP4.5 is a stabilized emission scenario with radiative forcing of approximately 4.5 W/m2 in the year 2100, and this value is never exceeded (Thomson et al., 2011; Van Vuuren et al., 2011). This scenario assumes that emission mitigation policies are implemented to limit emissions and radiative forcing. On the other hand, RCP8.5 is a very high emission scenario with radiative forcing of approximately 8.5 W/m2 in the year 2100. Although the global warming process under RCP4.5, which is based on a medium-low GHG emission pathway is relatively slow compared to higher GHG emissions (e.g., RCP8.5), many studies have suggested that the global warming climate under RCP4.5 exerts impacts on hydroclimatic phenomena (Chen et al., 2017; Donnelly et al., 2017; Kim et al., 2020). However, global warming impacts under different RCPs on the regional changes of hydroclimatic extremes are not simple. In this regard, the results based on two RCPs (RCP4.5 and RCP8.5) can provide useful information for identifying the impacts of global warming on hydroclimatic extremes from those expected under different RCPs. This implies the need for minimum mitigation strategies as well as adaptation plans according to the global warming induced by GHG emissions, even those under the relatively low-impact RCPs (e.g., RCP4.5).

*- Line 136: What is a central year for reference for MME at 0.48, 1.5, and 2℃?*

► Thank you for this comment. The central year, which is the median year during the 30-year period, is the first year in which the 30-year running temperature anomaly surpasses the target temperature

240   compared to the PI level. The target temperatures in this study for the 30-year reference and future periods are 0.48 ℃ and 1.5 and 2 ℃. We have clarified this point in the revised manuscript as follows:

: In this process, the individual 30-year periods and their central years (i.e., the median year of each period) are determined based on the temperature anomalies relative to the temperature of the PI period. All five GCMs reach specific warming levels in their central years and in the 30-year reference and future periods under both RCP4.5 and RCP8.5 (Table 3 and

245   Figure S1). Because the individual GCMs simulate the climate based on their own physical climate system processes, the warming phases of the GCMs are different even under the same emissions forcing. In this study, the central year of each period is the first year in which the 30-year running temperature anomaly surpasses the target temperature above the temperature of the PI period. The temperature anomalies targeted in this study are 0.48 ℃ for the reference period and 1.5 and 2 ℃ for the two future periods.

250

*- Line 137-138: How come the central year of PI is 1895? When your PI period is 1861-1890? Please check.*

► The correct central year of the PI period (1861-1890) is 1875. We revised this point in the manuscript.

: Unlike the temperature taken from the central year of the PI period (1875), the temperature anomalies are calculated for the entire period.

255

*- Line: 155: provide the full name of VIC.*

► We added the full definition of VIC.

: The bias-corrected GCMs are used as meteorological forcings to run the variable infiltration capacity (VIC) hydrological

260   model.

*- Keep observational datasets section after section 2.1*

► We have reorganized the contents and revised the manuscript.

265   *- Line 197: what are the reference period and two future periods? Is it for individual GCM or MME? Please clear to avoid confusion.*

► Thank you for this comment. The reference period and two future periods indicate the periods corresponding to warming targets of 0.48 ℃ and 1.5 ℃ (and 2.0 ℃) for the individual GCMs. We clarify this point in the revised manuscript as follows:

270   : For the changes in temperature extremes, the numbers of tropical days (TR), frost days (FD), warm nights (TN90P), and cold nights (TN10P) are calculated using daily minimum temperature data during the reference period and two future periods for each selected GCM, as shown in Table 3.

*- Table 4: Also, provide a source of indices (e.g., minimum temperature, maximum temperature, precipitation, and runoff) in*

275   *another column.*

► We provided the sources of the indices in the 4th column of Table 4.

**Table 4: Definitions of the hydroclimatic extreme indices using minimum temperature (denoted by TN), maximum temperature (denoted by TX), precipitation (denoted by PR), and runoff data, where 'i' and 'j' represent the month and year, respectively.**

| Index name (label) | Index definition | Unit | Source of indices |
|---|---|---|---|
| Tropical nights (TR) | The number of days when $TN_{ij} > 20\ ℃$ | Days | Minimum temperature |
| Frost days (FD) | The number of days when $TN_{ij} < 0\ ℃$ | Days | |
| Warm nights (TN90P) | The number of days when $TN_{ij} > TNref90$; here, TNref90 is the calendar day 90th percentile centered on a 5-day window for the reference period of individual GCMs | Days | |
| Cold nights (TN10P) | The number of days when $TN_{ij} < TNref10$; here, TNref10 is the calendar day 10th percentile centered on a 5-day window for the reference period of individual GCMs | Days | |
| Summer days (SU) | The number of days when $TX_{ij} > 25\ ℃$ | Days | Maximum temperature |
| Ice days (ID) | The number of days when $TX_{ij} < 0\ ℃$ | Days | |
| Warm days (TX90P) | The number of days when $TX_{ij} > TXref90$; here, TXref90 is the calendar day 90th percentile centered on a 5-day window for the reference period of individual GCMs | Days | |
| Cold days (TX10P) | The number of days when $TX_{ij} < TXref10$; here, TXref10 is the calendar day 10th percentile centered on a 5-day window for the reference period of individual GCMs | Days | |
| Very wet day precipitation (P95) | The total precipitation when $PR_{ij}$ exceeds the 95th percentile of the wet day precipitation in the reference period of individual GCMs | Mm | Precipitation |
| Extreme wet day precipitation (P99) | The total precipitation when $PR_{ij}$ exceeds the 99th percentile of the wet day precipitation in the reference period of individual GCMs | Mm | |
| Annual maximum precipitation (PX1D) | The maximum 1-day precipitation | Mm | |
| Maximum 2-day precipitation (PX2D) | The maximum consecutive 2-day precipitation | Mm | |
| Maximum 3-day precipitation (PX3D) | The maximum consecutive 3-day precipitation | Mm | |
| Maximum 5-day precipitation (PX5D) | The maximum consecutive 5-day precipitation | Mm | |
| Minimum 7-day runoff (DWF07) | The minimum consecutive 7-day runoff | Mm | Runoff |
| Minimum 30-day runoff (DWF30) | The minimum consecutive 30-day runoff | Mm | |
| Annual maximum runoff (MDF) | The maximum daily runoff | Mm | |

280

*- Section 3.1: Is climatic zone classification is based on observed data? And which year? Also, mention which observation reanalysis or APHRODITE and University Washington data?*

► We have clarified this point in the revised manuscript as follows:

285    : The climate zones over the Asian monsoon region in this study are classified based on long-term (30-year; 1976-2005) observation datasets (i.e., precipitation from APHRODITE; minimum and maximum temperatures from the University of Washington).

*- Line 219: "the bias-corrected GCMs are validated" Justify this result for temperature as well.*

290 ► Thank you for this comment. We have shown the validation results of the bias-corrected GCMs for temperature in Figure S5, and we have revised the manuscript accordingly (see response to the final comment No. 3)

295

300

*References (Added)*

305 Liu, J., Xu, H., Luo, J.J. and Deng, J.: Distinctive Evolutions of Eurasian Warming and Extreme Events Before and After Global Warming Would Stabilize at 1.5 °C, Earth's Future, 7, 151-161, 2019.

McSweeney, C.F. Jones, R.G., Lee, R.W. and Rowell, D.P: Selecting CMIP5 GCMs for downscaling over multiple regions, Climate Dynamics, 44, 3237-3260, 2015.

Tegegne, G., Melesse, A.M. and Worqlul, A.W.: Development of multi-model ensemble approach for enhanced assessment

310 of impacts of climate change on climate extremes, Science of The Total Environment, 704, 12321-12330, https://doi.org/10.1016/j.scitotenv.2019.135357, 2020.Yatagai, A., Kamiguchi, K., Arakawa, O., Hamada, A., Yasutomi, N. and Kitoh, A.: APHRODITE: Constracting a Long-Term Daily Gridded Precipitation Dataset for Asia Based on a Dense Network of Rain Gauges, Bulletin of the American Meteorological Society, 93, 1401-1415. https://doi:10.1175/BAMS-D-11-00122.1, 2012.

315 Thomas, V., Albert, J.R.G., and Perez, R.T.: Climate-Related Disasters in Asia and the Pacific, ADB Economics Working Paper Series No. 358, Asian Development Bank, Philippines, 38 pp., 2013.

Thomas, V., Albert, J.R.G., and Hepburn, C.: Contributors to the frequency of intense climate disasters in Asia-Pacific countries, Climatic Change, 126, 381-398, https://doi.org/10.1007/s10584-014-1232-y, 2014.

Thomas, V., and Lopez, R.: Global increase in Climate-Related Disasters, ADB Economics Working Paper Series No. 466,

320 Asian Development Bank, Philippines, 38 pp., 2015.

Xu, Y., Gao, X. and Giorgi, F.J.: Upgrades to the reliability ensemble averaging method for producing probabilistic climate-change projections, Climate Res., 41, 2375-2385, 2010.

Zhao, Y., Li, Z., Cai, S. and Wang, H.: Characteristics of extreme precipitation and runoff in the Xijiang River Basin at global warming of 1.5 °C and 2 °C, Natural Hazards, 101, 669-688, 2020.

325

**Reply to Reviewer (#2)'s Comments:**

*The manuscript presents an analysis of likely changes in future temperature, precipitation, and runoff extremes over different climatic regions of Asia. Climate projections are obtained from a suite of climate models and a hydrologic model is used to translate climate to runoff. While the manuscript addresses a topic of relevance to the journal, it has several major shortcomings that prohibit readers from interpreting the results and gauging their reliability. These are listed below:*

► We appreciate the reviewer's feedback and helpful comments. Kindly find our detailed response to each comment below.

*1. Novelty: There are several studies that discuss the consequences of 1.5 °C and 2.0 °C global warming on hydrology of river basins across the globe. A number of such analyses are cited in the paper but some significant research is not discussed. For example Betts et al. (2018) discuss the difference between the two global warming levels in terms of hydrologic extremes as well as food security. Similarly, Doll et al. (2018) employ two hydrologic models and an ensemble of bias corrected climate models to understand how freshwater related hazards are likely to change across the globe under the two warming levels. A number of similar studies can be found. The authors need to justify how their analysis adds value to this literature. A statistical analysis of expected changes is useful but these numbers need to be eventually translated into variables that have direct impact on society (such as food availability, flood hazard, etc.). Perhaps the authors can provide some policy relevant insights to the readers, for example, by suggesting how the adaptive measures will vary across different climate regions.*

► We fully agree with your valuable suggestion. As the reviewer suggested, we have added the related text in "Introduction" and "Section 4. Discussion and conclusions" to provide an in-depth survey of relevant research on hydroclimatic responses to global warming and to emphasize the meaningful contribution of this manuscript to the literature. We suggested the needs of different adaptive measures based on unique regional responses, especially those of the hydroclimatic extremes, to an additional 0.5 °C of global warming. We clarified these points in an organized way in manuscript (line 72-86 and line 437-524).

[revised manuscript text omitted]
 | ** | ** | ** | - | ** | ** | ** | ** | ** | ** | ** | ** | ** | ** | ** | ** | ** | * | * |

-20  -15  -10  -8  -6  -4  -2  0  2  4  6  8  10  15  20  (%)

405

*2. Methodology: There are a number of issues here:*

*- a. Climate data: the analysis involves a number of steps and the text is a little hard to follow in this regard. For example, to select the five GCMs, a comparison with a multimodel mean is carried out (Page 3, line 117). But why are GCMs selected on the basis of their performance w.r.t the ensemble mean? Why not directly compare the individual GCM performance with the observed data and select those that best represent the observed climate in the Asia region? It is later revealed that a bias-correction is also carried out on the climate data. Was this bias correction carried out before or after the shortlisting of the five GCMs? Overall, the sequence of methods is unclear and many methodological choices are not defended well in the text. Maybe a flowchart to guide the readers through the main steps will help.*

► We directly evaluated the performance of each individual GCM with the observed data and selected those that best represent the observed climate in Asia. However, to select the GCMs carefully under the assumption that the MME is similar to the observed data compared with each GCM (Xu et al., 2020; Tegegne et al., 2020), the lower-performing GCMs with relatively poor statistics compared with the MME are screened. Then, only the best-performing GCMs with better MME values are selected, and the MME is used only as a means of ranking the GCM. It is difficult to determine the lower-performing GCMs without a comparison between the individual GCMs and MME values because the comparison between each GCM and the observed data does not provide the information needed to exclude the GCMs. We have clarified this point in the revised manuscript.

: The spatial correlation coefficient (SCC) and root-mean-square error (RMSE) between the historical simulation fields derived from each GCM and the observed fields are calculated for each of the twelve relevant variables over the Asian monsoon region, as these statistics are commonly used to examine the performance of GCMs in the simulation of observed spatial climate features (IPCC, 2013; McSweeney et al., 2015). Next, we apply the MME-based scoring rule for the selection of GCMs (Nyunt et al., 2012) to exclude low-performing GCMs and identify only the best-performing GCMs using a relative concept because the scoring rule based on the observed data does not provide the information needed to screen the GCMs. Therefore, the individual GCM statistics (i.e., the SCC and RMSE) are judged by comparison with the MME statistic. The MME statistics are considered as criteria to score each GCM under the assumption that the MME is similar to the observed data compared with the output from only one GCM (Xu et al., 2020; Tegegne et al., 2020).

► For the second comment, bias correction method is carried out after selected the five GCMs. Raw GCM outputs are commonly used for GCM performance evaluations because GCM outputs are necessarily fitted to the observations after a bias correction. We added an explanation of this in the revised manuscript.

: The five of the raw GCMs of the Coupled Model Intercomparison Project Phase 5 (CMIP5; Taylor et al., 2012) are selected by applying a unique evaluation procedure (Kim et al., 2020). Then, a reference 30-year period and two future 30-year periods of individual GCM projections are defined under warming targets of 0.48, 1.5 and 2.0 ℃ above PI levels (1861-1890) based on a time sampling method. Then, these daily forcing data (e.g., precipitation, maximum temperature, and minimum temperature) are extracted from the five selected GCM projections and then statistically bias-corrected using the quantile mapping method. The bias-corrected GCMs are used as meteorological forcings to run the variable infiltration capacity (VIC) hydrological model.

► For the final comments, we clarified all these points and modified the manuscript (section 2.1~section 2.6) with a flowchart (Figure 2) to guide the readers through the main steps.

**: 2.3 Methodology**

Figure 2 presents a flowchart of the entire procedure used in the study. To simulate the climate during both historical and future periods, climate projections forced by historical and representative concentration pathways (RCPs) 4.5 and 8.5 are selected. The five of the raw GCMs of the Coupled Model Intercomparison Project Phase 5 (CMIP5; Taylor et al., 2012) are selected by applying a unique evaluation procedure (Kim et al., 2020). Then, a reference 30-year period and two future 30-year periods of individual GCM projections are defined under warming targets of 0.48, 1.5 and 2.0 ℃ above PI levels (1861-1890) based on a time sampling method. Then, these daily forcing data (e.g., precipitation, maximum temperature, and minimum temperature) are extracted from the five selected GCM projections and then statistically bias-corrected using the quantile mapping method. The bias-corrected GCMs are used as meteorological forcings to run the variable infiltration capacity (VIC) hydrological model. The future changes in the hydroclimatic mean and extremes corresponding to the conditions at warming targets of 1.5 and 2.0 ℃ are spatially analyzed according to the identified subregions based on climate zones. We focus on the hydroclimatic extreme responses to temperature, precipitation, and runoff variations under global warming targets (i.e., 1.5 and 2.0 ℃) using extreme indices. A more detailed description of each procedure is provided in section 2.4, section 2.5, and section 2.6.

[Figure]

**Figure 2: Flowchart of the entire procedure used in this study.**

*- b. Hydrologic model and runoff projections: the variable infiltration capacity model is a commonly used model to obtain continental to global scale runoff projections. However, many studies limit their analysis to understanding patterns in mean annual runoff. In this study, however, the focus is on runoff extremes, which are inherently harder to capture than a long-term mean value. However, the calibration and validation of the model are not included in the main manuscript. A regionalization approach is used to transfer parameters from gauged to ungauged sites, but how successful was this? It is important to show that the model with the regionalization scheme can capture the observed hydrologic extremes in the past. Only then, we can have some confidence regarding the reliability of the projections in the future.*

► Thank you for this comment. We understand the concern raised by the reviewer. Of course, it is inherently harder to simulate long-term mean runoff extremes using a hydrologic model. However, the results can aid in understanding runoff features when observed data are not available, even though the results are limited when simulating realistic runoff extremes. We have implemented an additional validation basins (i.e., total 20 basins suggested in Figure S2 and Table S2) and added validation of simulated extreme runoff by comparing measured extreme runoff based on the monthly maximum values (suggested in Figure S4) and the limitation when discussing the simulation of runoff extremes using the VIC model (line 236-255).

: To evaluate the reliability of the runoff results, the simulated mean and extreme runoff (i.e., monthly maximum runoff) values are validated by comparing with measured data. In this study, the simulated runoff is driven by observational meteorological forcings for the historical period (1950-2005) to compare the historical runoff records obtained from the Global Runoff Data Centre (GRDC). Some parameter validation results for the VIC model in 20 river basins (Figure S2) considering the data availability of measurement records are suggested in Table S2, Figure S3 and Figure S4, and additional results can be found in a previous study (Bae et al., 2013). The simulated monthly mean runoff obtained from the VIC model using observational meteorological input data shows high temporal correlation with the observed pattern for 6 basins (See Figure S3) and the range of correlation coefficients over the 20 basins are 0.58~0.97 (See Table S2). To evaluate the accuracy of the VIC model, we also consider other quantitative statistics, such as the model efficiency (ME), root-mean-square error (RMSE), and volume error (VE), as shown in Table S2. In general, simulated runoff qualitatively and quantitatively simulates the measured runoff. Figure S4 presents the scatter plot and box-whisker diagram of measured and

simulated monthly maximum runoff in the 20 basins. The assumptions used in parameter estimation and runoff analysis at the continental scale may impact the uncertainty in simulating monthly maximum runoff (See Figure S4a and Figure S4b), especially in capturing extreme runoff periods. Because it is inherently more difficult to simulate long-term mean runoff extremes using a hydrologic model, uncertainty exists between the simulated and measured extreme runoff data. Although simulated monthly maximum runoff (denoted as SIM) tends to underestimate the measured values (denoted as OBS), SIM commonly reproduces the OBS in terms of inter-quartile range (See Figure S4b) and the biases compared to variation range of OBS (See Figure S4c). The results can aid in understanding runoff features when observational data are not available, even though the results are limited when simulating realistic runoff. Overall, the validation results suggest that the VIC model is able to simulate mean and extreme runoff adequately.

490

495

[Figure]

**Figure S4: (a) Scatter plot of measured monthly maximum runoff and simulated monthly maximum runoff from the VIC model fed by observational meteorological input for all cases in the selected 20 basins (ALL: grey circles) and averages of all cases in each basin (AVE: blue circles) and (b) Box-whisker plot of measured monthly maximum runoff (denoted by OBS on the x-axis) and simulated monthly maximum runoff (denoted by SIM on the x-axis) in the 20 basins (unit: mm). (c) Box-whisker plot of the standard deviation of the observed monthly maximum runoff for all cases (denoted by OBS-STD on the x-axis) and the biases between OBS and SIM (denoted by BIAS on the x-axis).**

500

505

► We added a detailed description of the applied regionalization scheme and showed the scheme in the supporting information.

: We apply the hydrological regionalization method by transferring parameters obtained from gauged regions to ungauged regions based on the assumption that two basins with analogous climate features (e.g., based on the climate zone classification) exhibit similar hydrological responses. For runoff simulations at the global scale, Nijssen et al. (2001) obtained the parameters for an ungauged basin from the estimated parameters of a gauged basin with the same temperature and precipitation features. Xie et al. (2007) and Bae et al. (2013) employed the same approach leveraging climatological similarity according to Köppen's climate classification method and suggested the applicability of this method over China and Asia, respectively. In this study, both gauged basins and ungauged basins are divided into one of the climate zones to apply the hydrological regionalization method. We examine the optimal parameters for individual climate zones that effectively simulate runoff based on the estimated parameter sets obtained from all gauged basins within each climate zone. The optimal parameters of each climate zone are then transferred to the ungauged basins belonging to the same climate zone. In our previous study, the regionalization results were verified by assuming that some gauged basins are considered ungauged basins (Bae et al., 2013), and the results support the adaptability and applicability of the VIC model to simulate runoff throughout our study area.

510

515

520

► For the final comments, we added detailed description of the validation results for VIC model including the validation of simulated mean and extreme runoff in comparison with measured extreme

runoff (suggested in Table S2 and Figure S2-S4) and Table S2 Table S3 and line 236-255; (see response to first comment No. 2b)

*- c. Choice of extreme indices: The choice of indices for precipitation and runoff seem counter-intuitive. The precipitation indices focus on only high precipitation events while the runoff indices focus on both high and low events. Why not include precipitation extremes that involve minimum or very low precipitation?*

► We understand the concern raised by the reviewer. We set all daily precipitation amounts below 1.0 mm/day in the simulations to zero because GCMs tend to produce too little precipitation (<1 mm/day). Therefore, we did not include an analysis of the minimum or very low precipitation indices. In addition, for low precipitation extremes, although there is a duration-based concept (such as the dry spell length, which was suggested by the Expert Team on Climate Change Detection and Indices (ETCCDI)), this concept does not exactly match with low runoff events.

*- d. Selection of time periods: A time sampling method is used to identify the time period of analysis for various GCMs. The authors arrive at single time period for each GCM warming level. This suggests that a spatially aggregated value of climate indices was used to identify the time periods. However, the analysis focuses on different climate regions and it is possible that each climate region reaches a global warming level in different time periods. Why was this spatial heterogeneity ignored? The same applies on the bias correction methodology, which could have been applied on each homogeneous climate region one by one. On a similar note, the climate zone classification results are presented in Line 206 onwards. Is this classification carried out using observed data or GCM data? How sensitive is the classification to the choice of climate data?*

► Thank you for this comment. Of course, it is possible that each climate region does not present the same regional temperature increase as that occurring at the global scale. Therefore, globally aggregated warming targets do not necessarily mean that they can be universally acceptable because an increase in the global mean temperature does not translate into regional and local impacts in a straightforward manner (Knutti et al., 2015). In this regard, it is necessary to identify regionally emerging challenges faced by global warming targets because global warming levels above the preindustrial level are a global concept that is defined based on the globally aggregated mean temperature rather than the regional mean temperature. Based on these issues, we assessed the different regional hydroclimatic extreme climatic responses to global warming in this study.

► For the second comment, statistical bias correction (e.g., quantile mapping) methods adjust the simulated climate outputs by fitting the observed climate features. In general, the bias correction method is performed for each grid cell (or each point) data corresponding the gridded GCM outputs and observed data with the grid system. Therefore, we apply the quantile mapping method for each grid cell within the study domain.

► The climate classification is carried out using observed data for a long-term historical period (1976-2005). We clarify this point in the revised manuscript as follows:

560    : The climate zones over the Asian monsoon region in this study are classified based on long-term (30-year; 1976-2005) observation datasets (i.e., precipitation from APHRODITE; minimum and maximum temperatures from the University of Washington).

► For the final comment, the substantial differences exist among the individual observation datasets due to the analysis methodology such as the quality control of input data and spatial/temporal interpolation
565    used in producing these simulated datasets. Therefore, the observation datasets after the data quality management process (e.g., quality control, homogeneity testing) show similar spatial features though partly with biases (Tanarhte et al., 2012). Therefore, the classification of climate zone depends on the applied climate data (e.g., data sources, data periods). For instance, a level of uncertainty in the areas occupied by different Koppen climate type is smaller than 1% using the different period historical
570    dataset (Kalvová et al., 2003).

*Kalvová, J., Halenka, T., Bezpalcová, K. et al.: Köppen Climate Types in Observed and Simulated Climates, Studia Geophysica et Geodaetica 47, 185–202, 2003*

*Tanarhte, M., Hadjinicolaou, P. and Lelieveld, J.: Intercomparison of temperature and precipitation data sets based on observations in the Mediterranean and the Middle, Journal of Geophysical Research, 117, 1-12, 2012*

575

*- e. Choice of scenarios: it is not clear why RCP4.5 was chosen for the analysis when RCP6.0 and RCP8.5 are equally relevant.*

► Thank you for this comment. As the reviewer and editor suggested, we have implemented an additional RCP (i.e., RCP8.5); we have analyzed the results of changes in hydroclimatic extremes under 1.5 and 2 °C of global warming based on two different RCPs in the revise paper (i.e., RCP4.5 and RCP8.5).
580    In this regard, We have clarified the reason why we selected 2 RCPs as follows:

: Our focus is to understand the changes in extreme hydroclimatic conditions under global warming environments of 1.5 and 2.0 ℃. The timing to reach specific warming levels for individual GCMs depends on the representative concentration pathway  because future projections are forced by these scenarios. The temperature response to different RCPs varies, and therefore, the increasing trend and slope of the global mean temperature differ. Here, the analysis is based on RCP4.5 and
585    RCP8.5, which are commonly considered for realistic future projections. RCP4.5 is a stabilized emission scenario with radiative forcing of approximately 4.5 W/m2 in the year 2100, and this value is never exceeded (Thomson et al., 2011; Van Vuuren et al., 2011). This scenario assumes that emission mitigation policies are implemented to limit emissions and radiative forcing. On the other hand, RCP8.5 is a very high emission scenario with radiative forcing of approximately 8.5 W/m2 in the year 2100. Although the global warming process under RCP4.5, which is based on a medium-low GHG
590    emission pathway is relatively slow compared to higher GHG emissions (e.g., RCP8.5), many studies have suggested that the global warming climate under RCP4.5 exerts impacts on hydroclimatic phenomena (Chen et al., 2017; Donnelly et al., 2017; Kim et al., 2020). However, global warming impacts under different RCPs on the regional changes of hydroclimatic extremes are not simple. In this regard, the results based on two RCPs (RCP4.5 and RCP8.5) can provide useful information for identifying the impacts of global warming on hydroclimatic extremes from those expected under different RCPs. This
595    implies the need for minimum mitigation strategies as well as adaptation plans according to the global warming induced by GHG emissions, even those under the relatively low-impact RCPs (e.g., RCP4.5).

*3. Presentation: Overall, the manuscript can gain from improvement in language. In addition, the figure clarity can be improved. The figure captions are not very descriptive and it is hard to follow what is on the figures without carefully reading the main text. Please explain all symbols and abbreviations used in the figures in the caption itself.*

► The English grammar and expression have been polished by a professional agency.

► We upgraded the figures to high quality (600 dpi).

► We have thoroughly reviewed and modified the figure captions (including all symbols and abbreviations).

*References (Added)*

Liu, J., Xu, H., Luo, J.J. and Deng, J.: Distinctive Evolutions of Eurasian Warming and Extreme Events Before and After Global Warming Would Stabilize at 1.5 °C, Earth's Future, 7, 151-161, 2019.

McSweeney, C.F. Jones, R.G., Lee, R.W. and Rowell, D.P: Selecting CMIP5 GCMs for downscaling over multiple regions, Climate Dynamics, 44, 3237-3260, 2015.

Tegegne, G., Melesse, A.M. and Worqlul, A.W.: Development of multi-model ensemble approach for enhanced assessment of impacts of climate change on climate extremes, Science of The Total Environment, 704, 12321-12330, https://doi.org/10.1016/j.scitotenv.2019.135357, 2020.Yatagai, A., Kamiguchi, K., Arakawa, O., Hamada, A., Yasutomi, N. and Kitoh, A.: APHRODITE: Constracting a Long-Term Daily Gridded Precipitation Dataset for Asia Based on a Dense Network of Rain Gauges, Bulletin of the American Meteorological Society, 93, 1401-1415. https://doi:10.1175/BAMS-D-11-00122.1, 2012.

Thomas, V., Albert, J.R.G., and Perez, R.T.: Climate-Related Disasters in Asia and the Pacific, ADB Economics Working Paper Series No. 358, Asian Development Bank, Philippines, 38 pp., 2013.

Thomas, V., Albert, J.R.G., and Hepburn, C.: Contributors to the frequency of intense climate disasters in Asia-Pacific countries, Climatic Change, 126, 381-398, https://doi.org/10.1007/s10584-014-1232-y, 2014.

Thomas, V., and Lopez, R.: Global increase in Climate-Related Disasters, ADB Economics Working Paper Series No. 466, Asian Development Bank, Philippines, 38 pp., 2015.

Xu, Y., Gao, X. and Giorgi, F.J.: Upgrades to the reliability ensemble averaging method for producing probabilistic climate-change projections, Climate Res., 41, 2375-2385, 2010.

Zhao, Y., Li, Z., Cai, S. and Wang, H.: Characteristics of extreme precipitation and runoff in the Xijiang River Basin at global warming of 1.5 °C and 2 °C, Natural Hazards, 101, 669-688, 2020.

**Reply to Editor's Comments:**

*Thank you for your interesting manuscript and taking part in discussion round. Two referees have carefully reviewed your paper and ranked your work from fair to good. While they concur that the overall topic is interesting - they have also pointed*
645 *out number of limitations/issues. Both reviewers also kindly expressed their willingness to review the revised manuscript. I would invite you to revise the manuscript taking into account all the suggestions/comments of both reviewers into account.*

► We appreciate the review's and editor's feedback and helpful comments. Kindly find our detailed response to each comment below.

650 *As both reviewers have stated - please pay careful attention 1) to state the novelty of your study beyond what is already available in the research domain on possible consequences of 1.5 °C and 2.0 °C global warming including hydrometeorological extremes – or in words of the Reviewer #2 "to justify how their analysis adds value to this literature". 2) More careful consideration on the skill of the VIC model in simulating hydrologic extremes – employ a range of river basins in the study domain (than just six ones in Asia Monsoon region; available in the GRDC archive/other sources) to gain more*
655 *confidence in overall model based assessment results. And 3) also pointed by me earlier in preliminarily round and also echoed by the Reviewer – I strongly suggest the authors to implement other RCP scenarios (6.0/8.5) – and contrast their findings with respect to that of the RCP4.5.*

► Thank you for this comment. We fully agree with your valuable suggestion. We have added a discussion in "Section 4. Discussion and conclusions" to provide an in-depth survey of relevant
660 research on hydroclimatic responses to global warming and to emphasize the meaningful contribution of this manuscript to the literature. We have suggested the need for different adaptive measures based on unique regional responses, especially those of the hydroclimatic extremes, to an additional 0.5 °C of global warming. We have also clarified these points in an organized way in the manuscript (see response to Reviewer #2 first comment)

665 ► We understand the concern raised by the reviewer. To enhance the reliability of the runoff simulations, we have added more validation (i.e., total 20 basins suggested in Figure S2 and Table S2) of the simulated mean runoff and extreme runoff (e.g., monthly maximum values) by comparing the measured extreme runoff (Figure S4) with additional observational runoff data for river basins in the study domain. Also, we have added spatial distributions of the observation-driven runoff simulation
670 and MME-driven runoff simulation in terms of both annual mean runoff and daily maximum runoff for the entire domain in Figure S6. Finally, we have indicated the limitations of this work when discussing the simulation of runoff extremes using the VIC model. The added text is as follows:

: To evaluate the reliability of the runoff results, the simulated mean and extreme runoff (i.e., monthly maximum runoff) values are validated by comparing with measured data. In this study, the simulated runoff is driven by observational
675 meteorological forcings for the historical period (1950-2005) to compare the historical runoff records obtained from the Global Runoff Data Centre (GRDC). Some parameter validation results for the VIC model in 20 river basins (Figure S2) considering the data availability of measurement records are suggested in Table S2, Figure S3 and Figure S4, and additional results can be found in a previous study (Bae et al., 2013). The simulated monthly mean runoff obtained from the VIC model using observational meteorological input data shows high temporal correlation with the observed pattern for 6 basins

680  (See Figure S3) and the range of correlation coefficients over the 20 basins are 0.58~0.97 (See Table S2). To evaluate the

accuracy of the VIC model, we also consider other quantitative statistics, such as the model efficiency (ME), root-meansquare error (RMSE), and volume error (VE), as shown in Table S2. In general, simulated runoff qualitatively and

quantitatively simulates the measured runoff. Figure S4 presents the scatter plot and box-whisker diagram of measured and

simulated monthly maximum runoff in the 20 basins. The assumptions used in parameter estimation and runoff analysis at

685  the continental scale may impact the uncertainty in simulating monthly maximum runoff (See Figure S4a and Figure S4b),

especially in capturing extreme runoff periods. Because it is inherently more difficult to simulate long-term mean runoff

extremes using a hydrologic model, uncertainty exists between the simulated and measured extreme runoff data. Although

simulated monthly maximum runoff (denoted as SIM) tends to underestimate the measured values (denoted as OBS), SIM

commonly reproduces the OBS in terms of inter-quartile range (See Figure S4b) and the biases compared to variation range

690  of OBS (See Figure S4c). The results can aid in understanding runoff features when observational data are not available,

even though the results are limited when simulating realistic runoff. Overall, the validation results suggest that the VIC

model is able to simulate mean and extreme runoff adequately.

[Figure]

695  **Figure S4: (a) Scatter plot of measured monthly maximum runoff and simulated monthly maximum runoff from the VIC model fed by observational meteorological input for all cases in the selected 20 basins (ALL: grey circles) and averages of all cases in each basin (AVE: blue circles) and (b) Box-whisker plot of measured monthly maximum runoff (denoted by OBS on the x-axis) and simulated monthly maximum runoff (denoted by SIM on the x-axis) in the 20 basins (unit: mm). (c) Box-whisker plot of the standard deviation of the observed monthly maximum runoff for all cases (denoted by OBS-STD on the x-axis) and the biases between OBS and SIM (denoted by BIAS on the x-axis).**
700

► As the reviewer suggested, we have implemented an additional RCP (i.e., RCP8.5); we have analyzed

the results of changes in hydroclimatic extremes under 1.5 and 2.0 °C of global warming based on two

different RCPs in the revise paper (i.e., RCP4.5 and RCP8.5). Moreover, we have presented the results

705  and these points in an organized way in manuscript (in "Section 3. Results" and "Section 4. Discussion

and conclusions").

---

## Referee Report (RR1)

Authors have addressed my concerns in revised manuscript. I feel that manuscript is in good shape and could be accepted after minor revision. I have two concerns:

1. Kindly revise figure 3, figure S5 and figure S6. Plot should have OBS value and difference (in percentage for precipitation and runoff and in $^{o}C$ for temperature) between MME and OBS.

2. Quality of figures is not up to mark. I recommend to use Generic Mapping Tools (GMT) or R or other plotting software.

---

## Author Response (AR2)

Dear Editor and Reviewers:

First, we would like to thank the editor and reviewers for their helpful comments and suggestions, which improved the quality of our manuscript. We agree with most of the concerns raised by the reviewers and have therefore modified the manuscript according to the reviewers' comments and suggestions. Newly added and modified text is highlighted in yellow in the revised manuscript, and our point-by-point responses to the reviewers' comments are provided below. We hope that the revised manuscript is now suitable for publication in Hydrology and Earth System Sciences.

**Reply to Reviewer (#1)'s Comments:**

*Authors have addressed my concerns in revised manuscript. I feel that manuscript is in good shape and could be accepted after minor revision. I have two concerns:*

► We appreciate the reviewer's feedback and helpful comments. Kindly find our detailed response to each comment below.

*1. Kindly revise figure 3, figure S5 and figure S6. Plot should have OBS value and difference (in percentage for precipitation and runoff and in ᵒC for temperature) between MME and OBS.*

► We have revised the figures (Figure 3, Figure S5 and Figure S6) and added a related description in the manuscript.

: The percentage bias (hereafter referred to as BIAS) between the OBS and MME is calculated to examine the quantitative error in the MME. The MME properly captures both the spatial pattern and the magnitude of PANN and PX1D (Figure 3a, b). The relatively large magnitude of bias in PANN (PX1D) is shown in the region with low PANN (PX1D).

[Figure]

**Figure 3: Spatial distributions of the (a) annual mean precipitation (PANN) and (b) annual maximum precipitation (PX1D) for the historical period (1976-2005) in the Asian monsoon region derived from observations (OBS) and the MME of bias-corrected outputs from the five GCMs. BIAS (i.e., the 3rd column in each row) represents the percentage bias in PANN (PX1D) between OBS and MME.**

[Figure]

**Figure S5: Spatial distributions of the (a) annual minimum temperature (unit: °C) and (b) annual maximum temperature (unit: °C) for the historical period (1976-2005) in the Asian monsoon region. OBS and MME denote the values obtained from the observational temperature dataset and the MME of bias-corrected outputs from the five GCMs, respectively. BIAS (i.e., the 3rd column in each row) represents the percentage bias in individual variables between OBS and MME.**

[Figure]

**Figure S6: Spatial distributions of the (a) annual mean runoff (unit: mm) and (b) daily maximum runoff (unit: mm/day)**
**for the historical period (1976-2005) in the Asian monsoon region. OBS denotes the simulated runoff from the VIC**
**model fed by observational precipitation data (i.e., APHRODITE). MME denotes the simulated runoff from the VIC**
**model fed by the MME of the bias-corrected outputs from the five GCMs. BIAS (i.e., the 3rd column in each row)**
**represents the percentage bias in individual variables between OBS and MME.**

*2. Quality of figures is not up to mark. I recommend to use Generic Mapping Tools (GMT) or R or other plotting*
     *software.*

       ► Thank you for this comment. Although we used the general tool (and software) in this study

       (e.g., NCAR Command Language (NCL) and Grapher), the quality of figures were not

       enough due to the file format in the manuscript. We checked and submitted the figures with high quality as a ".zip" file (i.e., Individual image files in TIF format; 600 dpi).

**Reply to Reviewer (#2)'s Comments:**

*The revised manuscript has improved considerably in its presentation as well as description of methods and results. The authors have adequately addressed the main issues raised by the reviewers. The manuscript presents an interesting analysis of likely changes in hydro-climatic extremes considering different climate regions in the Asia monsoon region. The addition of flowchart is useful to understand the methodology. The paper can be considered for publication after minor improvements in presentation, a few are suggested below.*

► We appreciate the reviewer's feedback and helpful comments. Kindly find our detailed response to each comment below.

 *1. The abstract still reads quite poorly and needs to be improved. The abstract focuses on generalized insights while missing out on highlighting the regional differences, which is the main contribution of this paper. While the*

*last line says that the sensitivities are different, this difference is merely mentioned in the 2nd last line in reference to the cold and polar climates. In addition, the abstract has some unclear text. Some editorial suggestions:*

 ► We fully agree with your valuable suggestion. As the reviewer suggested, we have modified the abstract considering the raised suggestions, including the above comments from 1a to 1f, as follows:

: Understanding the influence of global warming on regional hydroclimatic extremes is challenging. To reduce the potential risk of extremes under future climate states, assessing the change in extreme climate events is important, especially in Asia, due to spatial variability of climate and its seasonal variability. Here, the changes in hydroclimatic extremes are assessed over the Asian monsoon region under global mean temperature warming targets of 1.5 and 2.0 ℃ above preindustrial levels based on representative concentration pathways (RCPs) 4.5 and 8.5. Analyses of the subregions classified using regional climate characteristics are performed based on the multimodel ensemble mean (MME) of five bias-corrected global climate models (GCMs). For runoff extremes, the hydrologic responses to 1.5 and 2.0 ℃ global warming targets are simulated based on the variable infiltration capacity (VIC) model. Changes in temperature extremes show increasing warm extremes and decreasing cold extremes in all climate zones with strong robustness under global warming conditions.

However, the hottest extreme temperatures occur more frequently in low-latitude regions with tropical climates. Changes in mean annual precipitation and mean annual runoff and low runoff extremes represent the large spatial variations with weak robustness based on intermodel agreements. Global warming is expected to consistently intensify maximum extreme precipitation events (usually exceeding a 10 % increase in intensity under 2.0 ℃ of warming) in all climate zones. The precipitation change patterns directly contribute to the spatial extent and magnitude of the high runoff extremes. Regardless of regional climate characteristics and RCPs, this behavior is expected to be enhanced under the 2.0 ℃ (compared with the 1.5 ℃) warming scenario and increase the likelihood of flood risk (up to 10 %). More importantly, an extra 0.5 ℃ of global warming under 2 RCPs will amplify the change in hydroclimatic extremes on temperature, precipitation and runoff with strong robustness, especially in cold (and polar) climate zones. The results of this study clearly show the consistent changes in regional hydroclimatic extremes related to temperature and high precipitation and suggest that hydroclimatic sensitivities can differ based on regional climate characteristics and type of extreme variables under warmer conditions over Asia.

*a. Consider rephrasing the first sentence in the abstract. Suggestion: 'Understanding the influence of global warming on regional hydro-climatic extremes is challenging.'*

► We have revised the abstract considering the reviewer's comment.

*b. Line 2: 'change of extreme' should be 'change in extreme'*

► We have revised the abstract considering the reviewer's comment.

*c. Line 3: 'due to various ....' can be 'due to spatial variability of climate and its seasonal variability'.*

► We have revised the abstract considering the reviewer's comment.

*d. Line 14: 'significant' is a statistical term, is a statistical significance implied here? I suggest to review the use of 'significant' and 'robustness' throughout to present a mathematically consistent meaning.*

► Thank you for this comment. We understand the concern raised by the reviewer. The level of agreement among the multiple projections, which is used to assess the robustness (or confidence) of climate projections (Tebaldi et al., 2011; Saeed et al., 2018), is suggested to provide a certain level of reliability in this study. Therefore, we have revised the abstract and added this point to the manuscript as follows:

: The level of agreement among the multiple projections is used to assess the robustness (or confidence) of climate projections (Tebaldi et al., 2011; Saeed et al., 2018).

: Tebaldi, C., Arblaster, J.M. and Knutti, R.: Mapping model agreement on future climate projections, Geophysical Research Letters, 38, L23701, 2011.

Saeed, F., Bethke, I., Fischer, E., Legutke, S., Shiogama, H., Stone, D.A. and Schleussner, C.-F.: Robust changes in tropical rainy season length at 1.5 ℃ and 2 ℃, Environmental Research Letter, 13, 064024, https://doi.org/10.1088/1748-9326/aab797, 2018.

*e. Lines 14-15: 'fewer than 45 days.... Fewer than 32 days' how are these related to warm and cold extremes, which should be captured on a temperature scale.*

► We have revised the abstract considering the reviewer's comment.

*f. Lines 15-16: 'changes in precipitation .... show' can be 'mean annual precipitation, mean annual runoff and low runoff extremes show'*

► We have revised the abstract considering the reviewer's comment.

*2. Line 74: remove 'the impact of' from 'the impact of global temperature...'*

► We revised this point in the manuscript (line 72).

*3. Line 74: replace 'separately' with 'differently'*

► We revised this point in the manuscript (line 72).

*4. Lines 75-76: 'hence, global ...' this line is redundant and repeats information from prior sentence. Please*
*restructure this part of the text.*

► Thank you for this comment. We fully agree with your valuable suggestion. We performed the suggested revision as follows:

: The hydroclimatic changes in response to global warming reflect unique regional responses because the global temperature increases impact each region differently due to changes in regional climate features. However, examining how different regional hydroclimatic extremes are caused by the impact of global warming remains challenging. To the best of our knowledge, relatively few studies have examined the impacts of global warming on extreme hydroclimatic variable-related responses considering the regional climate in Asia (Liu et al., 2019; Kim et al., 2020; Zhao et al., 2020).

*5. Lines 81-82: Remove 'since climate extremes ....component'*

► We revised this point in the manuscript (line 81).

*6. Line 235: 'gridded runoff'*

► We revised this point in the manuscript (line 236).

*7. Line 254-255: 'Overall, the validation ...' this claim is conflicting with the explanation prior that there is underestimation of observed maximum runoff values, though the inter-quartile range are captured. Perhaps this line is not needed and the readers can themselves decide how adequate are the simulations based on the observations reported earlier in this paragraph.*

► Thank you for this comment. We fully agree with your valuable suggestion. Therefore, we have removed this sentence in the manuscript (line 255).

*8. Line 328: what is 'not suggested'*

► We clarify this point in the revised manuscript as follows:

: These features (e.g., change patterns and spatial distributions) are shown in the results under RCP8.5 (related figure not suggested here).
Figure 5 shows the area-averaged changes in the cold and warm extreme indices derived from the results under RCP4.5 shown in Figure 4 (and under RCP8.5);

*9. Line 385: ')' without an opening round bracket*

► We revised this point in the manuscript.

*10. Figures 5, 7, 9, 11: can all be consolidated into a single table. Right now the number of figures for the manuscript is on the higher side.*

► Thank you for this comment. We understand the concern raised by the reviewer. However, we would like to keep the original figures as it is. If we consolidate these figures as a single table, it contains a large number of individual values (e.g., for individual climate zones, RCPs, extreme indices, global warming conditions). For this reason, figures could be more effective for identifying and comparing the different regional patterns of each climatic extreme.

*11. Section 3.2 and 3.3 mainly present several quantitative results. They are tedious to read with the numbers interspersed throughout (see for example, lines 400-405). Perhaps the text in these sections can be rewritten to focus on the important insights related to regional differences and differences between 1.5 and 2.0 degree warmings. A table can be used to consolidated the numbers being discussed.*

► We fully agree with your valuable suggestion. We have minimized the quantitative results and modified the text in "Sections 3.2 and 3.3". Therefore, we performed the suggested revision as follows:

: The change in FD over Asia represents the largest decrease of approximately -10.0 days at 1.5 ℃ of warming and -14.1 days at 2.0 ℃ of warming under the two RCPs. The change in ID also decreases by approximately -6.4 days at 1.5 ℃ of warming and -9.0 days at 2.0 ℃ of warming under the two RCPs. A large reduction in both FD and ID is detected in the cold climate zones (Ds, Dw, and Df) and polar climate zones (ET) with lower temperature records than the other climate zones. In contrast, the change in TR over Asia represents the largest increase of approximately 13.6 days (15.0 days) at 1.5 ℃ of warming and 20.6 days at 2.0 ℃ of warming under the two RCPs. Similarly, the change in SU is an increase of approximately 11.2 days at 1.5 ℃ of warming and 15.7 days at 2.0 ℃ of warming under the two RCPs. While the difference in the value of the results from the RCPs is the largest (i.e., approximately 1.4 days) in TR, it is similar in the other temperature extremes (i.e., FD, ID and SU).

: Warm days (TX90P) over Asia are projected to increase by 27.4 % under 2.0 ℃ of warming and by 18.7 % under 1.5 ℃ of warming for the two RCPs. Moreover, warm nights (TN90P) are projected to increase by 33.0 % under 2.0 ℃ of warming and by 23.6 % under 1.5 ℃ of warming under the two RCPs. The rate of warm days (TX90P) increase and warm nights (TN90P) increase are higher under RCP8.5 compared to RCP4.5. Conversely, cold days (TX10P) are projected to decrease by -7.4 % above PI levels on average in Asia at 2.0 ℃ of warming and by -6.1 % at 1.5 ℃ of warming under the two RCPs. Cold nights (TN10P) are projected to decrease by -8.3 % under 2.0 ℃ of warming and by -7.1 % under 1.5 ℃ of warming under the two RCPs. The rate of cold days (TX10P) decrease and cold nights (TN10P) decrease are slightly steeper under RCP8.5 than under RCP4.5. A large disparity between RCP4.5 and RCP8.5 is found in the change patterns of TX90P above the 50th percentile compared to TN90P. Overall, these change features in TN are more intense than those in TX (Figure 6a, c), which agrees with previous findings (IPCC, 2018).

: Figure 9 presents the area-averaged changes in annual mean precipitation (PANN) and PX1D compared to the REF period under 1.5 and 2.0 ℃ warming conditions based on RCP4.5 (RCP8.5). The changes in PX1D are greater than the changes in PANN in most climate zones except Bs and Bw (shown in Figure 8a and Figure S7a) under both RCP4.5 and RCP8.5. An increase in PANN under global warming based on the two RCPs compared with the REF period ranges from 0.1 % to 10.7 % at 1.5 ℃ of warming and from 11.7 % to 11.9 %

at 2.0 ℃ of warming. Similarly, under the two RCPs, PX1D is projected to significantly increase from 5.7 % to 11.2 % under 1.5 ℃ of warming and from 8.0 % to 15.2 % under 2.0 ℃ of warming. Namely, warming of 2.0 ℃ results in higher precipitation than warming of 1.5 ℃ in terms of both the PANN and PX1D irrespective of RCP scenarios.

*12. Line 426: remove 'changes in the', also please consider introducing all shorthands (MDF, RANN etc.) in the methods section. It is also difficult to follow the shorthands for climate zones, consider using the full form itself instead of the shorthand whenever possible.*

►  Thank you for this comment. We have made the suggested revision in the manuscript.

►  We have introduced all shorthands related to the extremes in the "Section 2.6 Extreme indices" and others in the relevant section. Additionally, we have attempted to suggest the full form with the shorthands in the manuscript whenever possible.

*13. Line 446-447: seems to be contradictory to the main claim in the abstract that there are considerable regional differences in climate sensitivities. Please correct the abstract to reflect the specific results. Use generalization*

*only when supported by the analysis. Instead use specific results such as those mentioned on line 466.*

►  We modified the abstract considering this point and performed the suggested revision in the manuscript (line 473).

*14. The text uses a confusing writing style. For example line 449-450 'changes in temperature ..... change*

*patterns'. What is a 'change pattern' and avoid using change twice in the same sentence. Please check the entire text to correct for this issue.*

►  We understand the concern raised by the reviewer. We have thoroughly reviewed and modified the expression.

*15. Figure 1: Perhaps it will be more interesting to visualize boundaries of large river basins that cover the region (as opposed to political boundaries). The major rivers in this region serve one of the largest populations in the world.*

►  Thank you for this comment. We fully agree with your valuable suggestion. Of course, it is more interesting and informative to visualize the boundaries of large river basins in Figure

1. However, the description of the classified climate zone can be more familiar to readers when it is based on the political boundaries rather than boundaries of large river basins. Additionally, considering the aspects of consistency with other figures and related descriptions, we would like to keep the original political boundary in Figure 1.

---

## Author Response (AR3)

Dear Editor:

First, we would like to thank the editor for their helpful comments and suggestions, which improved the quality of our manuscript. In the process of validation of revised paper, we were informed by an editorial support team that "Table 5" including the colors in the previous version of the manuscript (accepted by the Editor) will not be possible in the final version of the paper. If we want to adjust this, after the editor decision, the status changes to file upload, it can be acceptable that we upload the revised version to the system. Therefore, we have changed "Table 5" to "Figure 12" and uploaded the modified the manuscript. We hope that this manuscript is considered as a final version for publication in Hydrology and Earth System Sciences.

*Newly modified text is highlighted in yellow in the revised manuscript (i.e., "Section 4") as follows:*

[revised manuscript text omitted]